# DOGE-Train: Discrete Optimization on GPU with End-to-end Training

## Abstract

We present a fast, scalable, data-driven approach for solving linear relaxations of 0-1 integer linear programs using a graph neural network. Our solver is based on the Lagrange decomposition based algorithm [1]. We make the algorithm differentiable and perform backpropagation through the dual update scheme for end-to-end training of its algorithmic parameters. This allows to preserve the algorithm's theoretical properties including feasibility and guaranteed non-decrease in the lower bound. Since [1] can get stuck in suboptimal fixed points, we provide additional freedom to our graph neural network to predict non-parametric update steps for escaping such points while maintaining dual feasibility. For training of the graph neural network we use an unsupervised loss and perform experiments on large-scale real world datasets. We train on smaller problems and test on larger ones showing strong generalization performance with a graph neural network comprising only around $10k$ parameters. Our solver achieves significantly faster performance and better dual objectives than its non-learned version [1]. In comparison to commercial solvers our learned solver achieves close to optimal objective values of LP relaxations and is faster by up to an order of magnitude on very large problems from structured prediction and on selected combinatorial optimization problems. Our code will be made available upon acceptance.

## 1 Introduction

Integer linear programs (ILP) are a universal tool for solving combinatorial optimization problems. While great progress has been made on improving ILP solvers over the past several decades, some fundamental questions for future improvements remain open: Can ILP solvers make effective use of the massive parallelism afforded by GPUs and can modern machine learning meaningfully help? As of now the consensus seems that neither GPUs nor ML have yet helped general purpose ILP solvers in a fundamental way. In particular, this holds true for LP solvers which are a key component of most commonly used ILP approaches. LP solvers produce lower bounds on the optimal solution objective and are integral for many heuristics to decode feasible integral solutions. For many problems the ILP solvers spend most of the time on solving multiple LP relaxations, hence any impact GPUs and ML can have will directly translate into overall improvement of ILP solvers.

State of the art LP solvers [23, 13, 17, 4, 18] make little utility of modern machine learning but rather use either hand-designed or auto-tuned parameters and update rules. Moreover, with the exception of [18] these solvers are not open-source, hence researchers' ability to assess the potential of neural networks for improving LP solvers is limited. From a conceptual point of view traditional solver paradigms, e.g. simplex or interior point methods, are not GPU friendly and contain non-differentiable steps (such as pivot selection for simplex). Additionally, their high complexity further complicates

Submitted to 36th Conference on Neural Information Processing Systems (NeurIPS 2022). Do not distribute.

any effort at making them differentiable. This makes utilization of neural networks and GPUs for solver improvement difficult.

We propose a new way to use the potential of GPU parallelism and modern ML to obtain advances in LP relaxation solvers for ILPs. We argue that due to the difficulties in putting GPUs and ML to work in traditional solver methodologies, investigation of new paradigms is called for. To this end we build upon the recent work of [1] which proposed a massively parallel GPU friendly solver for 0-1 integer linear programming using Lagrange decomposition. The solver exhibits faster performance than traditional CPU solvers on large-scale problems making good use of GPU parallelism. Also due to its comparatively simple control flow and its usage of simple arithmetic operations for all its operations it can be made differentiable. This allows to train its parameters and predict update steps that will allow for faster convergence and overcoming fixed points from which the basic version of the algorithm suffers. This results in superior performance as compared to the non-learned version [1]. We obtain small gaps to (I)LP optima on a diverse range of large scale structured prediction problems, QAPLib [8] and independent set problems [39]. We are up to an order of magnitude faster than traditional ILP solvers.

**Contributions**   We propose to learn the Lagrange decomposition based algorithm [1] for solving LP relaxations of ILP problems and show its benefits. In particular,

- We make the dual update steps of [1] differentiable. This allows us to predict parameters of the update steps so that faster convergence is achieved as compared to using hand-picked values.
- We train a predictor for arbitrary non-parametric update steps that allow to escape suboptimal fixed points into which the parametric update steps of [1] can fall.
- We propose to train predictors for both the parametric and non-parametric updates in fully unsupervised manner. Our loss optimizes for parameters/update steps producing large improvements in the dual lower bound over a long time horizon.
- We show the benefits of our learned massively parallel GPU approach on a wide range of problems. We have chosen structured prediction tasks including graph matching [29] and cell tracking [24]. From theoretical computer science we compare on the QAPLib [8] dataset and randomly generated independent set problems [39].

## 2   Related Work

### 2.1   Learning to solve Combinatorial Optimization

ML has been used to improve various aspects of solving combinatorial problems. For the standard branch-and-cut ILP solvers the works [19, 22, 35] learn variable selection for branching. The approaches [14, 35] learn to fix a subset of integer variables in ILPs to their hopefully optimal values to improve finding high quality primal solutions. The works [43, 54] learn variable selection for the large neighborhood search heuristic for obtaining primal solutions to ILPs. Selecting good cuts through scoring them with neural networks was investigated in [26, 46]. While all these approaches result in runtime and solution quality improvements, only a few works tackle the important task of speeding up ILP relaxations by ML. Specifically, the work [11] used graph neural network (GNN) to predict variable orderings of decision diagrams representing combinatorial optimization problems. The goal is to obtain an ordering such that a corresponding dual lower bound is maximal. To our knowledge it is the only work that addresses computing ILP relaxations with ML. For constraint satisfaction problems [40, 9, 47] train GNN while [47] train in an unsupervised manner. For narrow subclasses of problems primal heuristics have been augmented through learning some of their decisions, e.g. for capacitated vehicle routing [36] and traveling salesman [55]. For a more complete overview of ML for combinatorial optimization we refer to the detailed surveys [6, 10].

### 2.2   Massively parallel combinatorial optimization

Massively parallel algorithms running on GPU have been proposed for narrow problem classes, including inference in [41, 56] and dense [45] Markov Random Fields, multicut [2] and for max-flow [49, 53]. The algorithm [1] on which our work is based is, to our knowledge, the only generic ILP solver that can make adequate use of parallelism offered by GPUs.

## 2.3 Unrolling algorithms for parameter learning

Algorithms containing differentiable iterative procedures are combined with neural networks for improving performance of such algorithms. One of the earliest works in this direction is [21] which embedded sparse coding algorithms in a neural network by unrolling. For solving inverse problems [57, 12] unroll through ADMM and non-linear diffusion resp. Overall, such approaches show more generalization power than pure neural networks based ones as shown in the survey [34]. Slightly different than from the above works, neural networks were used to predict update directions for training other neural networks (e.g. in [3]).

# 3 Method

We first recapitulate the Lagrange decomposition approach to binary ILPs from [31] and the deferred min-marginal averaging scheme for its solution proposed in [1]. We highlight possible parameters of the update steps which we will predict by training a graph neural network. Proofs are in the Appendix.

## 3.1 Lagrange Decomposition & Deferred Min-Marginal Averaging

**Definition 1** (Binary Program [31])**.** Let a linear objective $c \in \mathbb{R}^n$ and $m$ variable subsets $\mathcal{I}_j \subset [n]$ of constraints with feasible set $\mathcal{X}_j \subset \{0,1\}^{\mathcal{I}_j}$ for $j \in [m]$ be given. The corresponding binary program is

$$\min_{x \in \{0,1\}^n} \langle c, x \rangle \quad \text{s.t.} \quad x_{\mathcal{I}_j} \in \mathcal{X}_j \quad \forall j \in [m], \tag{BP}$$

where $x_{\mathcal{I}_j}$ is the restriction to variables in $\mathcal{I}_j$.

Any binary ILP $\min_{x \in \{0,1\}^n} \langle c, x \rangle$ s.t. $Ax \leq b$ where $A \in \mathbb{R}^{m \times n}$ can be written as (BP) by associating each constraint $a_j^T x \leq b_j$ for $j \in [m]$ with its own subproblem $\mathcal{X}_j$.

In order to obtain a problem formulation amenable for parallel optimization we consider its Lagrange dual which decomposes the full problem (BP) into a series of coupled subproblems.

**Definition 2** (Lagrangean dual problem [31])**.** Define the set of subproblems that constrain variable $i$ as $\mathcal{J}_i = \{j \in [m] \mid i \in \mathcal{I}_j\}$. Let the energy for subproblem $j \in [m]$ w.r.t. Lagrangean dual variables $\lambda_{\bullet j} = (\lambda_{ij})_{i \in \mathcal{I}_j} \in \mathbb{R}^{\mathcal{I}_j}$ be

$$E^j(\lambda_{\bullet j}) = \min_{x \in \mathcal{X}_j} \langle \lambda_{\bullet j}, x \rangle. \tag{1}$$

Then the Lagrangean dual problem is defined as

$$\max_{\lambda} \sum_{j \in [m]} E^j(\lambda_{\bullet j}) \quad \text{s.t.} \quad \sum_{j \in \mathcal{J}_i} \lambda_{ij} = c_i \quad \forall i \in [n]. \tag{D}$$

The authors in [1] have proposed a parallelization friendly iterative algorithm for updating Lagrange multipliers $\lambda$ for maximizing (D), see Algorithm 1. We write it in a slightly adapted form since it will allow us to easily describe its backpropagation. The algorithm assigns the Lagrange variables in $u$-many disjoint blocks $B_1, \ldots, B_u$ in such a way that each block contains at most one Lagrange variable from each subproblem and all variables within a block are updated in parallel. The dual update scheme relies on computing min-marginal differences i.e., the difference of subproblem objectives when a certain variable is set to $1$ minus its objective when the same variable is set to $0$, see line 10 in Algorithm 1. These min-marginal differences are averaged out across subproblems via updates to Lagrange variables in line 11 in Algorithm 1. The crucial ingredient allowing parallelization is that in the min-marginal averaging step values from the last iteration are used (i.e. $M^{\text{in}}$), making synchronization between subproblems unnecessary.

In [1] the min-marginal averaging parameters of Algorithm 1 were set as $\omega = 0.5$ and $\alpha_{ij} = {}^1\!/_{|\mathcal{J}_i|}$ leading to uniform averaging. We generalize the min-marginal update step by considering more general parametric update steps. We allow $\omega \in (0, 1)$ and $\alpha$-values to be arbitrary convex combinations. In the next section we will show how to train these values to achieve faster convergence.

**Proposition 1** (Dual Feasibility and Monotonicity of Min-marginal Averaging)**.** *For any $\alpha_{ij} \geq 0$ with $\sum_{j \in \mathcal{J}_i} \alpha_{ij} = 1$ and $\omega_{ij} \in [0, 1]$ the min-marginal averaging step in line 11 in Algorithm 1 retains dual feasibility and is non-decreasing in the dual lower bound.*

---

**Algorithm 1:** Parallel Deferred Min-Marginal Averaging [1]

---

**Input:** Lagrange variables $\lambda_{ij} \, \forall i \in [n], j \in \mathcal{J}_i$, damping factors $\omega_{ij} \in (0,1) \, \forall i \in [n], j \in \mathcal{J}_i$, anisotropic min-marginal averaging weights $\alpha_{ij} \in (0,1) \, \forall i \in [n], j \in \mathcal{J}_i$, max. number of iterations $T$.

1  Initialize deferred min-marginal diff. $M = \mathbb{0}$
2  **for** $T$ *iterations* **do**
3     **for** *block* $B \in (B_1, \ldots B_u)$ **do**
4        $\lambda, M \leftarrow$ BlockUpdate $(B, \lambda, M, \alpha, \omega)$
5     **for** *block* $B \in (B_u, \ldots B_1)$ **do**
6        $\lambda, M \leftarrow$ BlockUpdate $(B, \lambda, M, \alpha, \omega)$
7  **return** $\lambda, M$
8  **Procedure** BlockUpdate$(B, \lambda^{\texttt{in}}, M^{\texttt{in}}, \alpha, \omega)$
9     **for** $ij \in B$ *in parallel* **do**
10       Compute $M_{ij}^{\texttt{out}} = \omega_{ij}[\min_{x \in \mathcal{X}_j : x_i = 1} \langle \lambda_{\bullet j}^{\texttt{in}}, x \rangle - \min_{x \in \mathcal{X}_j : x_i = 0} \langle \lambda_{\bullet j}^{\texttt{in}}, x \rangle]$
11       Update $\lambda_{ij}^{\texttt{out}} = \lambda_{ij}^{\texttt{in}} - M_{ij}^{\texttt{out}} + \alpha_{ij} \sum_{k \in \mathcal{J}_i} M_{ik}^{\texttt{in}}$
12    **return** $\lambda^{\texttt{out}}, M^{\texttt{out}}$

---

## 3.2 Backpropagation through Deferred Min-Marginal Averaging

We show below how to differentiate through Algorithm 1 with respect to the parameters $\alpha$ and $\omega$. This will ultimately allow us to learn these parameters such that faster convergence is achieved. To this end we describe backpropagation for a block update (lines 8- 12) of Alg. 1. All other operations can be tackled by automatic differentiation. For a block $B$ in $\{B_1, \ldots, B_u\}$ we view the Lagrangean update as a mapping $\mathcal{H} : (\mathbb{R}^{|B|})^4 \to (\mathbb{R}^{|B|})^2$, $(\lambda^{\texttt{in}}, M^{\texttt{in}}, \alpha, \omega) \mapsto (\lambda^{\texttt{out}}, M^{\texttt{out}})$.

Given a loss function $\mathcal{L} : \mathbb{R}^N \to \mathbb{R}$ we denote $\partial \mathcal{L} / \partial x$ by $\dot{x}$. Algorithm 2 shows backpropagation through $\mathcal{H}$ to compute the gradients $\dot{\lambda}^{\texttt{in}}, \dot{M}^{\texttt{in}}, \dot{\alpha}$ and $\dot{\omega}$.

**Proposition 2.** *Algorithm 2 performs backpropagation through $\mathcal{H}$.*

**Efficient Implementation** Generally, the naive computation of min-marginal differences and its backpropagation are both expensive operations as they require solving two optimization problems for each dual variable. In [1, 31] the authors represented each subproblem using binary decision diagrams (BDDs) for fast incremental computation of min-marginal differences. Their algorithm results in a computation graph involving only elementary arithmetic operations and taking minima over several variables. Using this computational graph we can implement the abstract Algorithm 2 efficiently and parallelize on GPU. For details we refer to the Appendix.

---

**Algorithm 2:** BlockUpdate backpropagation

---

**Input:** Forward pass inputs: $B, \lambda^{\texttt{in}}, M^{\texttt{in}}, \alpha, \omega$, gradients of forward pass output: $\dot{\lambda}^{\texttt{out}}, \dot{M}^{\texttt{out}}$, gradients of parameters $\dot{\alpha}, \dot{\omega}$

1  **for** $ij \in B$ *in parallel* **do**
2     $\dot{M}_{ij}^{\texttt{in}} = \sum_{k \in \mathcal{J}_i} \dot{\lambda}_{ik}^{\texttt{out}} \alpha_{ik}, \quad \dot{M}_{ij}^{\texttt{out}} = \dot{M}_{ij}^{\texttt{out}} - \dot{\lambda}_{ij}^{\texttt{out}}$
3     $\dot{\alpha}_{ij} = \dot{\alpha}_{ij} + \dot{\lambda}_{ij} \sum_{k \in \mathcal{J}_i} M_{ik}^{\texttt{in}}, \quad \dot{\omega}_{ij} = \dot{\omega}_{ij} + \dot{M}_{ij}^{\texttt{out}}[M_{ij}^{\texttt{out}}/\omega_{ij}]$
4     Compute minimizers $s^j(i, \beta) = \arg\min_{x \in \mathcal{X}_j : x_i = \beta} \langle \lambda_{\bullet j}^{\texttt{in}}, x \rangle, \, \forall \beta \in \{0, 1\}$
5     $\dot{\lambda}_{pj}^{\texttt{in}} = \dot{\lambda}_{pj}^{\texttt{out}} + \dot{M}_{ij}^{\texttt{out}} \omega_{ij}[s_p^j(i, 1) - s_p^j(i, 0)], \, \forall p \in \mathcal{I}_j$
6  **return** $\dot{\lambda}^{\texttt{in}}, \dot{M}^{\texttt{in}}, \dot{\alpha}, \dot{\omega}$

---

## 3.3 Non-Parametric Update Steps

Although the min-marginal averaging scheme of Alg. 1 guarantees non-decreasing lower bound, it can get stuck in suboptimal fixed points, see [50] for a discussion for the special case of MAP-MRF. To alleviate this shortcoming we allow arbitrary updates to Lagrange variables through a vector

$\theta \in \mathbb{R}^{|\lambda|}$ as

$$\lambda_{ij} \leftarrow \lambda_{ij} + \theta_{ij} - \frac{1}{|\mathcal{J}_i|} \sum_{k \in \mathcal{J}_i} \theta_{ik}, \ \forall i \in [n], j \in \mathcal{J}_i \tag{2}$$

where the last term ensures feasibility of updated Lagrange variables w.r.t. the dual problem (D).

## 3.4 Graph neural network

We train a graph neural network (GNN) to predict the parameters $\alpha, \omega$ of Alg. 1 and also the non-parametric update $\theta$ for (2). To this end we encode the dual problem (D) on a bipartite graph $\mathcal{G} = (\mathcal{V}, \mathcal{E})$. Its nodes correspond to primal variables $\mathcal{I}$ and subproblems $\mathcal{J}$ i.e., $\mathcal{V} = \mathcal{I} \cup \mathcal{J}$ and edges $\mathcal{E} = \{ij \mid i \in \mathcal{I}, j \in \mathcal{J}_i\}$ correspond to Lagrange multipliers. We need to predict values of $\alpha_{ij}, \omega_{ij}$ and $\theta_{ij}$ for each edge $ij$ in $\mathcal{E}$. We associate features $f = (f_{\mathcal{I}}, f_{\mathcal{J}}, f_{\mathcal{E}})$ with each entity of the graph which capture the current state of Alg. 1. Additionally, we encode a number of quantities as features which can make learning easier. For example, a history of previous dual objectives for each subproblem is encoded in the constraint nodes and minimizers of each subproblem (which correspond to a subgradient of the dual problem (D)) are encoded in the edge features $f_{\mathcal{E}}$. A complete list of features is provided in the Appendix.

**Message passing**   To perform message passing we use the transformer based graph convolution scheme of [42]. We first compute an embedding of all subproblems $j$ in $\mathcal{J}$ by receiving messages from adjacent nodes and edges as

$$\text{CONV}_{\mathcal{J}}(f_{\mathcal{I}}, f_{\mathcal{J}}, f_{\mathcal{E}}, \mathcal{E})_j = \mathbf{W_s} f_j + \sum_{i \mid ij \in \mathcal{E}} a_{ij}(f_j, f_{\mathcal{I}}, f_{\mathcal{E}}; \mathbf{W_a}) \left[\mathbf{W_t} f_i + \mathbf{W_e} f_{ij}\right], \tag{3}$$

where $\mathbf{W} = (\mathbf{W_a}, \mathbf{W_s}, \mathbf{W_t}, \mathbf{W_e})$ are trainable parameters and $a_{ij}(f_j, f_{\mathcal{I}}, f_{\mathcal{E}}; \mathbf{W_a})$ is the softmax attention weight between nodes $i$ and $j$ parameterized by $\mathbf{W_a}$. Afterwards we perform message passing in the reverse direction to compute embeddings for primal variables $\mathcal{I}$. Similar strategy for message passing on a bipartite graph was followed by [19].

**Recurrent connections**   Our default GNN as mentioned above only uses hand-crafted features to maintain a history of previous optimization rounds. To learn a summary of the past updates we optionally allow recurrent connections through an LSTM with forget gate [20]. The LSTM is only applied on primal variable nodes $\mathcal{I}$ and maintains cell states $s_{\mathcal{I}}$ which can be updated and used for parameter prediction in subsequent optimization rounds.

**Prediction**   The learned embeddings from GNN, LSTM outputs and solver features from Alg. 1 are consumed by a multi-layer perceptron $\Phi$ to predict the required variables for each edge $ij$ in $\mathcal{E}$. Afterwards we transform these outputs so that they satisfy Prop. 1.

The exact sequence of operations performed by the graph neural network are shown in Alg. 3 where $[u_1, \ldots, u_k]$ denotes concatenation of vectors $u_1, \ldots, u_k$, LN denotes layer normalization [5] and $\text{LSTM}_{\mathcal{I}}$ stands for an LSTM cell which operates on each primal variable node.

---

**Algorithm 3:** Parameter prediction by GNN

**Input:** Primal variable features $f_{\mathcal{I}}$ and cell states $s_{\mathcal{I}}$, Subproblem features $f_{\mathcal{J}}$, Dual variable (edge) features $f_{\mathcal{E}}$, Set of edges $\mathcal{E}$.

1 $h_{\mathcal{J}} = \texttt{ReLU}\left(\texttt{LN}\left(\texttt{CONV}_{\mathcal{J}}\left(f_{\mathcal{I}}, f_{\mathcal{J}}, f_{\mathcal{E}}, \mathcal{E}\right)\right)\right)$      // Compute subproblems embeddings
2 $h_{\mathcal{I}} = \texttt{ReLU}\left(\texttt{LN}\left(\texttt{CONV}_{\mathcal{I}}\left(f_{\mathcal{I}}, [f_{\mathcal{J}}, h_{\mathcal{J}}], f_{\mathcal{E}}, \mathcal{E}\right)\right)\right)$    // Compute primal variable embeddings
3 $z_{\mathcal{I}}, s_{\mathcal{I}} = \texttt{LSTM}_{\mathcal{I}}(h_{\mathcal{I}}, s_{\mathcal{I}})$               // Compute output and cell state
4 $(\hat{\alpha}, \hat{\omega}, \theta) = \Phi\left([f_{\mathcal{I}}, h_{\mathcal{I}}, z_{\mathcal{I}}], [f_{\mathcal{J}}, h_{\mathcal{J}}], f_{\mathcal{E}}, \mathcal{E}\right)$    // Prediction per edge
5 $\alpha_{i\bullet} = \texttt{Softmax}(\hat{\alpha}_{i\bullet}), \forall i \in \mathcal{I}, \ \omega = \texttt{Sigmoid}(\hat{\omega})$ // Ensure non-decreasing obj., Prop. 1
6 **return** $\alpha, \omega, \theta, s_{\mathcal{I}}$

---

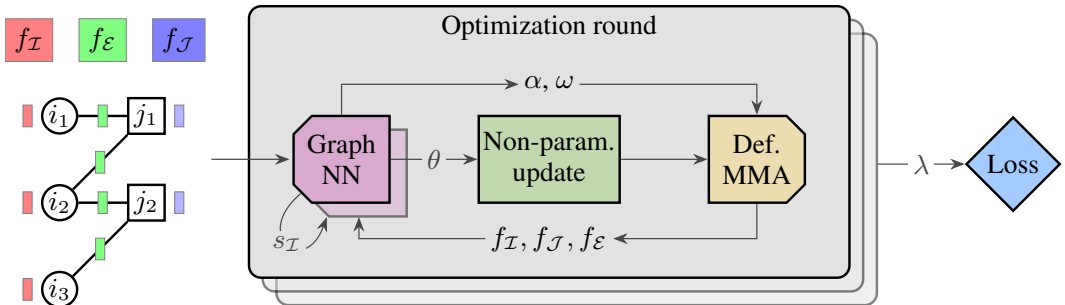

Figure 1: Our pipeline for optimizing the Lagrangean dual (D). The problem is encoded on a bipartite graph containing features $f_\mathcal{I}$, $f_\mathcal{J}$ and $f_\mathcal{E}$ for primal variables, subproblems and dual variables resp. A graph neural network (GNN) predicts the non-parameteric update $\theta$ (2) and parameters $\alpha$ and $\omega$ for Alg. 1. In one optimization round current set of Lagrange multipliers $\lambda$ are first updated by the non-parametric update using $\theta$. Afterwards deferred min-marginal averaging is performed parameterized by $\alpha$ and $\omega$. The updated solver features $f$ (which also includes $\lambda$) and LSTM cell states $s_\mathcal{I}$ are sent to the GNN in next optimization round. These rounds are repeated at most $R$-times during training and until convergence during inference.

## 3.5 Loss

Given the Lagrange variables $\lambda$ we directly use the dual objective (D) as an unsupervised loss to train the GNN. Thus, we maximize the loss $L$ defined as

$$\mathcal{L}(\lambda) = \sum_{j \in [m]} E^j(\lambda_{\bullet j}). \tag{4}$$

For a mini-batch of instances during training we take the mean of corresponding per-instance losses. For backpropagation, gradient of loss $\mathcal{L}$ w.r.t. Lagrange variables of a subproblem $j$ is computed by finding a minimizing assignment for that subproblem, written as

$$\left(\frac{\partial \mathcal{L}}{\partial \lambda}\right)_{\bullet j} = \mathrm{argmin}_{x \in \mathcal{X}_j} \langle \lambda_{\bullet j}, x \rangle \in \{0,1\}^{\mathcal{I}_j}. \tag{5}$$

The above gradient is then sent as input for backpropagation. For computing the minimizing assignment efficiently we use binary decision diagram representation of each subproblem as in [1, 31].

## 3.6 Overall pipeline

Our overall pipeline combining all building blocks from the previous sections is shown in Figure 1. We train our pipeline which contains multiple dual optimization rounds in a fashion similar to that of recurrent neural networks. One round of our dual optimization consists of message passing by GNN, a non-parametric update step and $T$ iterations of deferred min-marginal averaging. For computational efficiency we run our pipeline for at most $R$ dual optimization rounds during training. On each mini-batch we randomly sample a number of optimization rounds $r$ in $[R]$, run $r-1$ rounds without tracking gradients and backpropagate through the last round by computing the loss (4). For the pipeline with recurrent connections we backpropagate through last 3 rounds and apply the loss after each of these rounds. Since the task of dual optimization is relatively easier in early rounds as compared to later ones (where [1] can get stuck) we use two neural networks. The early stage network is trained if the randomly sampled $r$ is in $[0, R/2]$ and the late stage network is chosen otherwise. During testing we switch to the later stage network when the relative improvement in the dual objective by the early stage network becomes less than $10^{-6}$.

## 4 Experiments

As main evaluation metric we report convergence plots of the relative dual gap $g(t) \in [0,1]$ at time $t$

$$g(t) = \min\left(\frac{d^* - d(t)}{d^* - d_{init}}, 1.0\right) \tag{6}$$

where $d(t)$ is the dual objective at time $t$, $d^*$ is the optimal (or best known) objective value of the Lagrange relaxation (D) and $d_{init}$ is the objective value before optimization as computed by [1]. Additionally we also report per dataset averages of relative dual gap integral $g_I = \int g(t)dt$ [7], best objective value ($E$) and time taken ($t$) to obtain best objective. To cater the dominating effect of worse initial lower bounds on $g_I$ (as $g(t)$ can be close to 1 at $t \approx 0$) we start calculating $g_{\mathcal{I}}$ after a few rounds of our solver are completed. This start time is then also used to evaluate other algorithms for a fair comparison. To evaluate CPU solvers we use an AMD EPYC 7702 CPU. For the GPU solvers we use either one NVIDIA RTX 8000 (48GB) or A100 (80GB) GPU depending on instance size.

## 4.1 Algorithms

`Gurobi`: Results of the dual simplex algorithm from the commercial ILP solver [23].

`FastDOG`: The non-learned baseline [1] of Alg. 1 with $\omega_{ij} = 0.5$ and $\alpha_{ij} = 1/|\mathcal{J}_i|$.

`DOGE`: Our approach where we learn to predict parametric and non-parametric updates by using two graph neural networks for early and late-stage optimization. Size of the learned embeddings $h$ computed by the GNN in Alg. 3 is set to 16 for nodes and 8 for edges. For computing attention weights in (3) we use only one attention head for efficiency. The predictor $\Phi$ in Alg. 3 contains 4 linear layers with the ReLU activation. We train the networks using the Adam optimizer [30]. To prevent gradient overflow we use gradient clipping on model parameters by an $l^2$ norm of 50. The number of trainable parameters is $8k$.

`DOGE-M`: Variant of our method where we additionally use recurrent connections using LSTM. The cell state vector $s_i$ for each primal variable node $i \in \mathcal{I}$ has a size of 16. The number of trainable parameters is $12k$.

We have not tested against specialized heuristics for our benchmark problems since [1] has shown them to be on par or outperformed by `FastDOG`. For training our approach we use the frameworks [15, 16, 38] and implement the Algorithms 1,2 in CUDA [37] using [25, 28].

## 4.2 Datasets

*Cell tracking (CT)*: Instances of developing flywing tissue from cell tracking challenge [48] processed by [24] and obtained from [44]. We use the largest and hardest 3 instances, train on the 2 smaller instances and test on the largest one.

*Graph matching (GM)*: Instances of graph matching for matching nuclei in 3D microscopic images [32] processed by [29] and made publicly available through [44]. We train on 10 instances and test on the remaining 20 instances.

*Independent set (IS)*: Random instances of independent set problem generated using [39]. For training we generate 240 instances with $10k$ vertices each and test on 60 instances with $50k$ vertices. We generating edges between vertices in the graph with a probability of 0.25.

*QAPLib*: The benchmark dataset for quadratic assignment problems used in the combinatorial optimization community [8]. We train on 61 instances having up to 40 nodes and test on 35 instances having up to 70 nodes.

For each dataset we use a separate set of hyperparameters due to varying instance sizes given in Table 1. All our test datasets on average contain more than a million edges (i.e., Lagrange variables) while training instances are considerably smaller. For efficiency, during evaluation we use a larger value of $T$ in Alg. 1 than during training. For the *CT* dataset containing we learn only the non-parametric update steps (2) and fix the parameters in Alg. 1 to their default values from [1]. Learning these parameters gave slightly worse training loss at convergence.

## 4.3 Ablation study

We perform an ablation study to test the importance of various components of our approach. Starting from [1] as a baseline we first predict all parameters $\alpha, \omega, \theta$ through the two multi-layer perceptrons $\Phi$ for early and late stage optimization without using GNN. Next, we report results of using one network (instead of two) which is trained and tested for both early and later rounds of dual optimization. Lastly, we aim to seek the importance of learning parameters of Alg. 2 and the non-parametric update (2). To this end, we learn to predict only the non-parametric update and apply the loss directly on updated

Table 1: Hyperparameters of our approach and dataset statistics. $|\mathcal{I}| + |\mathcal{J}|$: Average number of variables and constraints in each dataset (# vertices in GNN); $\sum_{j=1}^{m} |\mathcal{J}_i|$: Average number of Lagrange multipliers (# edges in GNN); $T$: Number of iterations of Alg. 1 in each optimization round; $R$: max. number of training rounds; # itr. train: Number of training iterations.

| Dataset | $|\mathcal{I}| + |\mathcal{J}|$ $(\times 10^6)$ | | $\sum_{j=1}^{n} |\mathcal{J}_i|$ $(\times 10^6)$ | | $T$ | | $R$ | batch size | learn. rate | # itr. train | train time [hrs] |
|---|---|---|---|---|---|---|---|---|---|---|---|
| | train | test | train | test | train | test | | | | | |
| *CT* | 3.7 | 12.4 | 8.5 | 28 | 1 | 100 | 400 | 1 | 1e-3 | 500 | 14 |
| *GM* | 1.7 | 1.7 | 3.3 | 3.3 | 20 | 200 | 20 | 2 | 1e-3 | 400 | 4 |
| *IS* | 0.05 | 0.4 | 0.1 | 1.2 | 20 | 50 | 20 | 8 | 1e-3 | 2500 | 10 |
| *QAPLib* | 0.1 | 2.8 | 0.5 | 11 | 5 | 20 | 500 | 4 | 1e-3 | 1600 | 48 |

Table 2: Ablation study results on the *Graph matching* dataset. w/o GNN: Use only the two predictors $\Phi$ without GNN for early and late stage optimization; same network: use one network (GNN, $\Phi$) for both early and late stage; only non-param., param.: predict only the non-parametric update (2) or the parametric update (Alg. 1); w/o $\alpha$, $\omega$: does not predict $\alpha$ or $\omega$ resp.

| | w/o learn. ([1]) | w/o GNN | same network | only non-param. | only param. | w/o $\alpha$ | w/o $\omega$ | DOGE | DOGE-M |
|---|---|---|---|---|---|---|---|---|---|
| $g_I$ ($\downarrow$) | 21 | 0.42 | 0.95 | 2.3 | 0.7 | 0.36 | 0.35 | 0.33 | **0.19** |
| $E$ ($\uparrow$) | $-48912$ | $-48440$ | $-48444$ | $-48476$ | $-48444$ | $-48439$ | $-48439$ | $-48439$ | **$-48436$** |
| $t[s]$ ($\downarrow$) | 61 | 29 | 24 | 51 | 74 | 30 | 30 | 17 | 21 |

$\lambda$ without requiring backpropagation through Alg. 1. We also try learning a subset of parameters i.e., not predicting averaging weights $\alpha$ or damping factors $\omega$. Lastly, we report results of DOGE-M which uses recurrent connections. The results are in Table 2.

Firstly, from our ablation study we observe that learning even one of the two types of updates i.e., non-parametric or parametric already gives better results than the non-learned solver [1]. This is because non-parametric update can help in escaping fixed-points of [1] when they occur and the parametric update can help Alg. 1 in avoiding such fixed-points. Combining both of these strategies further improves the results. Secondly, we observe that performing message passing with GNN gives improvement over only using the predictor $\Phi$. Thirdly, we find using separate networks for early and late stage optimization gives better performance than using the same network for all stages. Lastly, using recurrent connections gives the best performance.

## 4.4 Results

Convergence plots of relative dual gaps change w.r.t. wall clock times are given in Figure 2. Rest of the evaluation metrics are reported in Table 3. For further details we refer to the Appendix.

**Discussion** As compared to the non-learned baseline FastDOG we reach an order of magnitude more accurate relaxation solutions, almost closing the gap to optimum as computed by Gurobi. We retain high speed afforded by exploiting GPU parallelism. Interestingly, we can often outperform FastDOG also in the early stage where optimization is easy. Our LSTM version DOGE-M has shown improved performance than the non-LSTM version. Especially it shows much improvement on the most difficult *QAPLib* dataset. On *QAPLib* Gurobi does not converge on instances with more than

Table 3: Results comparison on all datasets where the values are averaged within a dataset. Numbers in bold highlight the best performance.

| | *Cell tracking* | | | *Graph matching* | | | *Independent set* | | | *QAPLib* | | |
|---|---|---|---|---|---|---|---|---|---|---|---|---|
| | $g_I$ | $E(\times 10^8)$ | $t[s]$ | $g_I$ | $E(\times 10^4)$ | $t[s]$ | $g_I$ | $E(\times 10^8)$ | $t[s]$ | $g_I$ | $E(\times 10^6)$ | $t[s]$ |
| Gurobi [23] | 18 | **$-3.852$** | 809 | 9 | **$-4.8433$** | 278 | 14 | **$-2.4457$** | 52 | 3472 | 0.9 | 2618 |
| FastDOG [1] | 7 | $-3.863$ | 1005 | 21 | $-4.8912$ | 61 | 42 | $-2.4913$ | 9 | 276 | 5.7 | 1680 |
| DOGE | 2.4 | $-3.854$ | 1015 | 0.3 | $-4.8439$ | 17 | 0.3 | $-2.4460$ | 8 | 320 | 12.1 | 720 |
| DOGE-M | **2.1** | $-3.854$ | 730 | **0.2** | $-4.8436$ | 21 | **0.2** | $-24459$ | 5 | **131** | **14.5** | 861 |

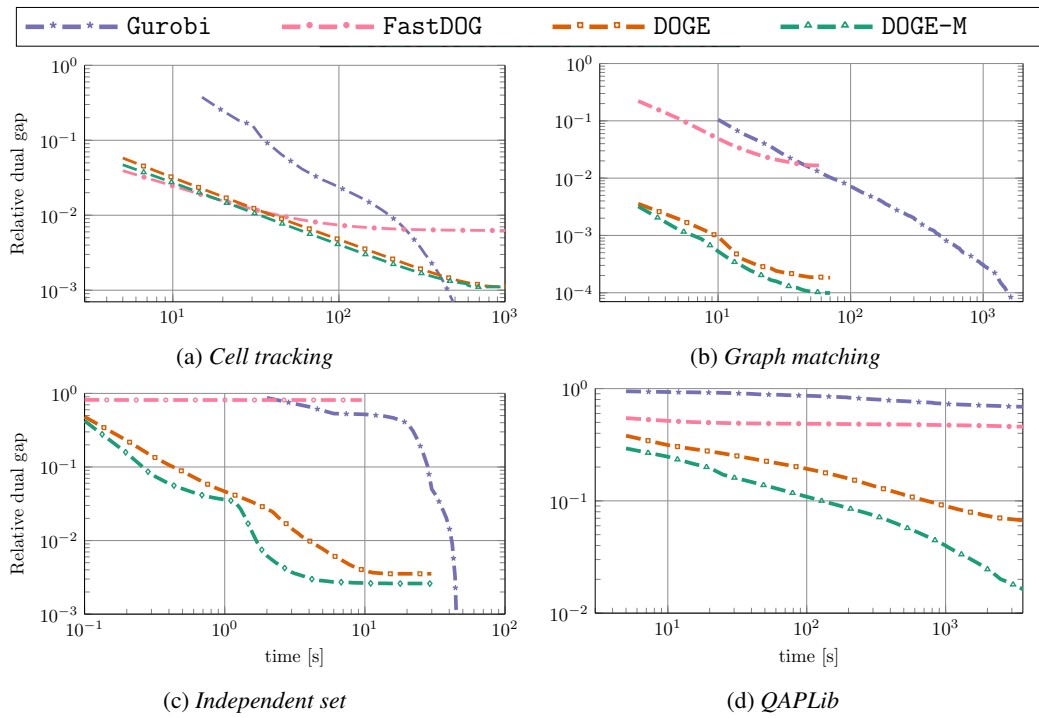

(a) *Cell tracking*

(b) *Graph matching*

(c) *Independent set*

(d) *QAPLib*

Figure 2: Convergence plots for $g(t)$ defined in (6), the relative dual gap to the optimum (or maximum suboptimal objective among all methods) of the relaxation (D). Both axes are logarithmic.

40 nodes within the time limit of one hour. We show convergence plots for smaller instances in the Appendix. The difference to `Gurobi` is most pronounced w.r.t. anytime performance measured by $g_I$, since our solver reaches good solutions relatively early.

**Limitations** While our approach gives solutions of high accuracy for the presented datasets, we have also tried our approach on other datasets (small cell tracking instances, MRFs for protein folding [27] and shape matching [51, 52]) where we were not able to obtain significant improvements w.r.t. the non-learned baseline [1]. For small cell tracking instances `FastDOG` already found the optimum in a moderate number of iterations, making it hard to beat. On shape matching and protein folding the parallelization of `FastDOG` did not bring enough speed-ups due to few large subproblems resulting in sequential bottlenecks. This limited the number of training iterations we could perform within a reasonable time.

We have set some hyperparameters in a dataset-dependent way. This was partly necessitated due to problem sizes e.g., training on long time horizons was not possible with very large instances. Moreover, these instances only permitted a limited number of parameters in our neural networks.

## 5 Conclusion

We have proposed a learning approach for solving relaxations to combinatorial optimization problems by backpropagating through and learning parameters for the non-learned baseline [1]. We demonstrated its potential in obtaining close to optimal solutions faster than with traditional methods.

Our work raises interesting follow-up questions: (i) Contrary to many approaches for backpropagation which replace non-smooth operations with smoothed variants (e.g. [33]) we directly compute (sub-) gradients for the non-smooth solver updates. Can smoothing of the solver help obtain a better backpropagated supervision? (ii) We argue that predicting good update steps for our solver is in itself an interesting and challenging problem for GNNs. We hope that our work can become a testbed for GNN architectures. (iii) There are a few desiderata for future learned solvers, including training universal models that generalize across different problem classes. Possibly more powerful GNNs and more involved training regimes are needed for this.

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
