# Appendix

## A  Proofs

### A.1  Proof of Proposition 1

The proof is an adaptation of the corresponding proof for $\omega_{ij} = 0.5$ and $\alpha_{ij} = \frac{1}{|\mathcal{J}_i|}$ given in [1].

*Proof.*
**Feasibility of iterates.** We prove

$$\sum_{j \in \mathcal{J}_i} \lambda_i^j + M_{ik} = c_i \tag{7}$$

just after line 4 and 6 in Algorithm 1. We do an inductive proof over the number of iterates w.r.t iterations $t$.

$t = 0$:  • After 4: Follows from $M = 0$ in line 1.
  • After 6: Let $\lambda'$, $M'$, be the values that are used as input to line 6 and $\lambda$ and $M$ be the ones returned in line 6. It holds that

$$\sum_{j \in \mathcal{J}_i} [\lambda_{ij} + M_{ij}] = \sum_{j \in \mathcal{J}_i} \left[ \lambda'_{ij} - M_{ij}(t) + \alpha_{ij} \sum_{k \in \mathcal{J}_i} (M'_{ik}) + M_{ij} \right] \tag{8}$$

$$= \sum_{j \in \mathcal{J}_i} \left[ \lambda'_{ij} + M'_{ij} \right] \tag{9}$$

$$= c_i . \tag{10}$$

by the proved inequality on $\lambda'$, $M'$ and the assumption that $\sum_{k \in \mathcal{J}_i} \alpha_{ij} = 1$.

$t > 0$:  Analoguously to the second point for $t = 0$.

**Non-decreasing Lower Bound.** In order to prove that iterates have non-decreasing lower bound we will consider an equivalent lifted representation in which proving the non-decreasing lower bound will be easier.

**Lifted Representation.** Introduce $\lambda_{ij}^\beta$ for $\beta \in \{0, 1\}$ and the subproblems

$$E(\lambda_{\bullet j}^1, \lambda_{\bullet j}^0) = \min_{x \in \mathcal{X}_j} x^\top \lambda_{\bullet j}^1 + (1 - x)^\top \lambda_{\bullet j}^0 \tag{11}$$

Then (D) is equivalent to

$$\max_{\lambda^1, \lambda^0} \sum_{j \in \mathcal{J}} E(\lambda_{\bullet j}^1, \lambda_{\bullet j}^0) \text{ s.t. } \sum_{j \in \mathcal{J}_i} \lambda_{ij}^\beta = \beta \cdot c_i \tag{12}$$

We have the transformation from original to lifted $\lambda$

$$\lambda \mapsto (\lambda^1 \leftarrow \lambda, \lambda^0 \leftarrow \mathbb{0}) \tag{13}$$

and from lifted to original $\lambda$ (except a constant term)

$$(\lambda^1, \lambda^0) \mapsto \lambda^1 - \lambda^0 . \tag{14}$$

It can be easily shown that the lower bounds are invariant under the above mappings and feasible $\lambda$ for (D) are mapped to feasible ones for (12) and vice versa.

The update rule line 11 in Algorithm 1 for the lifted representation can be written as

$$\lambda_{ij}^\beta \leftarrow \lambda_{ij}^\beta - \max((2\beta - 1)M_{ij}^{out}, 0) + \alpha_{ij} \cdot \sum_{jk \in \mathcal{J}_i} \min((2\beta - 1)M_{ik}^{in}, 0) \tag{15}$$

It can be easily shown that (15) and line 11 in Algorithm 1 are corresponding to each other under the transformation from lifted to original $\lambda$.

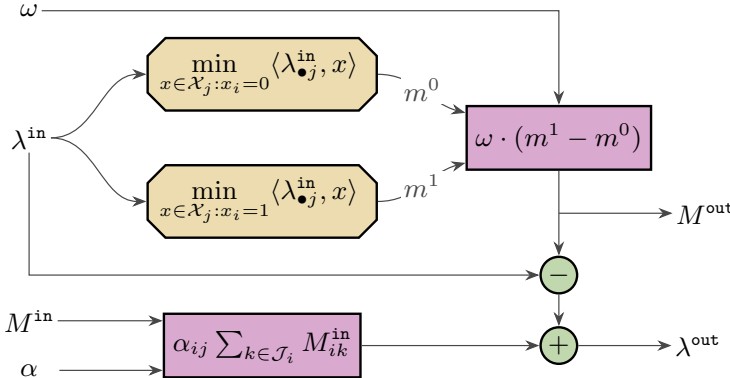

Figure 3: Computational graph of `BlockUpdate` in Alg. 1

**Continuation of Non-decreasing Lower Bound** Define

$$\lambda_{ij}'^{\beta} = \lambda_{ij} - \omega_{ij} \cdot \max((2\beta - 1)(\min_{x \in \mathcal{X}_j : x_j = \beta} \langle \lambda_{ij}^{in}, x \rangle - \min_{x \in \mathcal{X}_j : x_i = 1-\beta} \langle \lambda_{ij}^{in}, x \rangle), 0). \tag{16}$$

Then $E(\lambda'^{j,1}, \lambda'^{j,0}) = E(\lambda^{j,1}, \lambda^{j,0})$ are equal due to $\omega_{ij} \in [0, 1]$. Define next

$$\lambda_{ij}''^{\beta} = \lambda_{ij}' + \alpha_{ij} \sum_{k \in \mathcal{J}_i} \max((2\beta - 1)M_{ik}^{in}, 0). \tag{17}$$

Then $E(\lambda''^{j,1}, \lambda''^{j,0}) \geq E(\lambda'^{j,1}, \lambda'^{j,0})$ since $\lambda'' \geq \lambda'$ elementwise. This proves the claim. $\qquad\square$

## A.2 Proof of Proposition 2

*Proof.* The computational graph of `BlockUpdate` in Alg. 1 is shown in Figure 3. Assuming gradients $\partial \mathcal{L}/\partial M^{out}$ and $\partial \mathcal{L}/\partial \lambda^{out}$ are given. We first focus on lower part of Figure 3. By applying chain rule gradient of $M_{ij}^{in} \, \forall ij \in B$ is computed as

$$\frac{\partial \mathcal{L}}{\partial M_{ij}^{in}} = \sum_{p \in \mathcal{I}} \sum_{k \in \mathcal{J}_p} \frac{\partial \mathcal{L}}{\partial \lambda_{pk}^{out}} \frac{\partial \lambda_{pk}^{out}}{\partial M_{ij}^{in}} = \sum_{k \in \mathcal{J}_i} \frac{\partial \mathcal{L}}{\partial \lambda_{ik}^{out}} \frac{\partial \lambda_{ik}^{out}}{\partial M_{ij}^{in}} = \sum_{k \in \mathcal{J}_i} \frac{\partial \mathcal{L}}{\partial \lambda_{ik}^{out}} \alpha_{ij}. \tag{18}$$

Similarly gradient for $\alpha_{ij} \, \forall ij \in B$ is

$$\frac{\partial \mathcal{L}}{\partial \alpha_{ij}} = \sum_{p \in \mathcal{I}} \sum_{k \in \mathcal{J}_p} \frac{\partial \mathcal{L}}{\partial \lambda_{pk}^{out}} \frac{\partial \lambda_{pk}^{out}}{\partial \alpha_{ij}} = \frac{\partial \mathcal{L}}{\partial \lambda_{ij}^{out}} \frac{\partial \lambda_{ij}^{out}}{\partial \alpha_{ij}} = \frac{\partial \mathcal{L}}{\partial \lambda_{ij}^{out}} \sum_{k \in \mathcal{J}_i} M_{ik}^{in}, \tag{19}$$

Since we allow running Alg. 1 for more than one iteration with same parameters $(\alpha, \omega)$, the above gradient (19) is accumulated to existing gradients of $\alpha$ to obtain the result given by Alg. 2.

For the upper part of Figure 3 we first backpropagate gradients of $\lambda^{out}$ to $M^{out}$ to account for subtraction $(-)$ as

$$\frac{\partial \mathcal{L}}{\partial M^{out}} = \frac{\partial \mathcal{L}}{\partial M^{out}} - \frac{\partial \mathcal{L}}{\partial \lambda^{out}}. \tag{20}$$

Then the gradient w.r.t. damping factors $\omega_{ij} \, \forall ij \in B$ is

$$\frac{\partial \mathcal{L}}{\partial \omega_{ij}} = \frac{\partial \mathcal{L}}{\partial M_{ij}^{out}} \frac{\partial M_{ij}^{out}}{\partial \omega_{ij}} = \frac{\partial \mathcal{L}}{\partial M_{ij}^{out}} \left( m_{ij}^1 - m_{ij}^0 \right) = \frac{\partial \mathcal{L}}{\partial M_{ij}^{out}} \left( \frac{M_{ij}^{out}}{\omega_{ij}} \right), \tag{21}$$

which also needs to be accumulated to existing gradient as done for gradients of $\alpha$.

Lastly to backpropagate gradients to $\lambda^{in}$ we first calculate

$$\frac{\partial \mathcal{L}}{\partial m_{ij}^0} = \frac{\partial \mathcal{L}}{\partial M_{ij}^{out}} \frac{\partial M_{ij}^{out}}{\partial m_{ij}^0} = -\frac{\partial \mathcal{L}}{\partial M_{ij}^{out}} \omega_{ij}, \tag{22a}$$

$$\frac{\partial \mathcal{L}}{\partial m_{ij}^1} = \frac{\partial \mathcal{L}}{\partial M_{ij}^{out}} \frac{\partial M_{ij}^{out}}{\partial m_{ij}^1} = \frac{\partial \mathcal{L}}{\partial M_{ij}^{out}} \omega_{ij}. \tag{22b}$$

Then (sub-)gradient of min-marginals $m_{ij}^0, m_{ij}^1 \forall ij \in B$ w.r.t. $\lambda^{\mathtt{in}}$ are

$$\frac{\partial m_{ij}^\beta}{\partial \lambda} = \frac{\partial m_{ij}^\beta}{\partial \lambda_{\bullet j}} = \mathrm{argmin}_{x \in \mathcal{X}_j : x_{ij} = \beta} \langle \lambda_{\bullet j}, x \rangle, \quad \forall \beta \in \{0, 1\}. \tag{23}$$

Using the above relations (22), (23) and applying chain rule we obtain

$$\frac{\partial \mathcal{L}}{\partial \lambda_{ij}^{\mathtt{in}}} = \frac{\partial \mathcal{L}}{\partial \lambda_{ij}^{\mathtt{out}}} + \sum_{\beta \in \{0,1\}} \sum_{p \in \mathcal{I}} \sum_{k \in \mathcal{J}_p} \frac{\partial \mathcal{L}}{\partial m_{pk}^\beta} \frac{\partial m_{pk}^\beta}{\lambda_{ij}^{\mathtt{in}}} \tag{24a}$$

$$= \frac{\partial \mathcal{L}}{\partial \lambda_{ij}^{\mathtt{out}}} + \sum_{\beta \in \{0,1\}} \sum_{p \in \mathcal{I}_j} \frac{\partial \mathcal{L}}{\partial m_{pj}^\beta} \frac{\partial m_{pj}^\beta}{\lambda_{ij}^{\mathtt{in}}}, \forall ij \in B. \tag{24b}$$

$\square$

## B   Efficient min-marginal computation and backpropagation

Algorithms 1 and 2 in abstract terms require solving the subproblems each time a min-marginal value (or its gradient) is required. To make these procedures more efficient we represent each subproblem as binary decision diagrams (BDD) as done in [1]. We give a short overview below and refer to [1] for more details.

**Binary decision diagrams (BDD).**   A BDD is a directed acyclic graph with arc set $A$ starting at a root node $r$ and ending at two nodes $\top$ and $\bot$. For each variable $i$ the BDD contains one or more nodes in a set $\mathcal{P}_i$ where all $r\top$ paths pass through exactly one node in $\mathcal{P}_i$. All $r\top$ paths in the BDD correspond to feasible assignments of its corresponding subproblem. Lagrange variables of the subproblem can be used as weights in BDD arcs allowing also to calculate cost of these $r\top$ paths. This is done by creating two outgoing arcs for a node $v$ (except $\top, \bot$) in the BDD: a zero arc $vs^0(v)$ and a one arc $vs^1(v)$. If an $r\top$ path passes through zero arc $vs^0(v)$ it indicates that the corresponding variable has an assignment of 0 and 1 otherwise.

Therefore to compute the cost of assigning a 1 to variable $i$ one needs to check all $r\top$ paths which make use of the one arcs from all nodes in $\mathcal{P}_i$. In [1] the authors compute min-marginals by maintaining shortest path distances. Each node $v$ in the BDD maintains the cost of shortest path from root node $r$ (denoted by $\mathrm{SP}(r, v)$) and cost of shortest path to $\top$ node. These path costs are updated in `BlockUpdate` routine of Alg. 1. Min-marginals $m^0, m^1$ for a variable $i$ in subproblem $j$ can be computed efficiently as

$$m_{ij}^\beta = \min_{\substack{vs^\beta(v) \in A \\ v \in \mathcal{P}_i}} \left[ \mathrm{SP}(r, v) + \beta \cdot \lambda_{ij} + \mathrm{SP}(s^\beta(v), \top) \right]. \tag{25}$$

Backpropagation through min-marginals $m^0$, $m^1$ can then be done by finding the $\mathrm{argmin}$ in (25) instead of the $\min$ operation. Afterwards the gradients can be passed to Lagrange variables $\lambda$ and shortest path costs $\mathrm{SP}(r, \cdot)$, $\mathrm{SP}(\cdot, \top)$ which minimize (25). Since shortest path costs are also computed by $\min$ operations (see Alg. 3, 4 in [1]), gradients of these path costs can subsequently be backpropagated to the Lagrange variables by the $\mathrm{argmin}$ operation.

## C   Neural network details

### C.1   Hand-crafted features

The features used as input to the neural networks at every optimization round are provided in Table 4.

## D   Results

### D.1   Results on smaller instances of *QAPLib*

In Figure 4 we provide additional convergence plot calculated only on smaller instances of *QAPLib* dataset. These instances contain on average 1.6 million dual variables (instead of the overall test

Table 4: Features used for learning. Exponentially averaged features are computed with a smoothing factor of $0.9$. Features corresponding to the ILP remain fixed (i.e. node degrees, constraint type, $c$, $A$, $b$) whereas the remaining features are updated after every optimization round.

| Types | Feature description |
|---|---|
| Primal variables $f_{\mathcal{I}}$ | Normalized cost vector $c/\|c\|_{\infty}$ 
 Node degree ($|\mathcal{J}_i| \, \forall i \in \mathcal{I}$) |
| Subproblems $f_{\mathcal{J}}$ | Node degree ($|\mathcal{I}_j| \, \forall j \in \mathcal{J}$) 
 RHS vector $b$ in constraints $Ax \leq b$ 
 Indicator for constraint type ($\leq$ or $=$) 
 Current objective value per subproblem $[E^1(\bullet_j), \ldots, E^m(\lambda_{\bullet j})]$ 
 Exp. moving avg. of first, second order change in obj. value 
 Change in objective value due to last non-parametric update (2) |
| Dual variables $f_{\mathcal{E}}$ | Current optimal assignment of each subproblem 
 Exp. moving avg. of optimal assignment 
 Coefficients of constraint matrix $A$ 
 Current (normalized) Lagrange variables $\lambda/\|\lambda + M + \epsilon\|$ 
 Current (normalized) deferred min-marginal differences $M/\|\lambda + M + \epsilon\|$ |

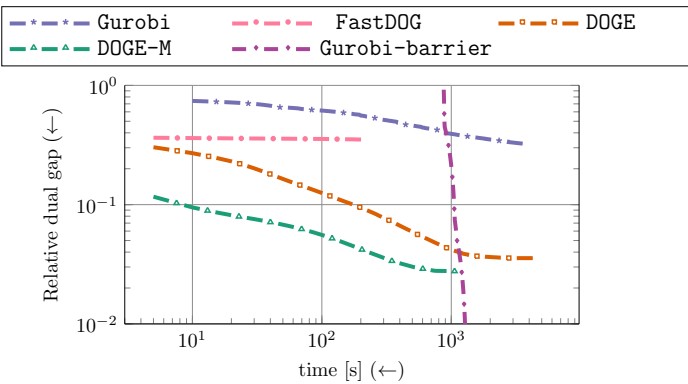

Figure 4: Convergence plots of smaller test instances of *QAPLib* ($\leq 40$ nodes).

split with 11 million). We observe that on relatively smaller instances our solvers `DOGE`, `DOGE-M` are surpassed by barrier method but not by dual simplex method of `Gurobi`. However, on larger instances barrier method could not perform any iteration within 1 hour timelimit.

### D.2 *Cell tracking*

Table 5: Detailed results on *Cell tracking* dataset. Until termination criteria contain results where we stop our solvers early w.r.t. relative improvement. These results are averaged and reported in Table 3. Best until max. itr.: We run our solver for at most 50000 iterations and report best results (so $R = 500$, $T = 100$).

| instance | method | Until termination criteria | | | | Best until max. num itr. | | | |
|---|---|---|---|---|---|---|---|---|---|
| | | $E$ ($\uparrow$) | $g(t)$ ($\downarrow$) | $t$ ($\downarrow$) | # itr. | $E$ ($\uparrow$) | $g(t)$ ($\downarrow$) | $t$ ($\downarrow$) | # itr. |
| | Gurobi | - | - | - | - | $-385235600$ | $0$ | $809$ | - |
| flywing-245 | DOGE | $-385424704$ | $0.00108$ | $1380$ | $50000$ | $-385424704$ | $0.00108$ | $1380$ | $50000$ |
| | DOGE-M | $-385428640$ | $0.00111$ | $730$ | $28900$ | $-385428544$ | $0.00111$ | $760$ | $30100$ |

### D.3 *Graph matching*

Table 6: Detailed results on *Graph matching* dataset. Until termination criteria contain results where we stop our solvers early w.r.t. relative improvement. These results are averaged and reported in Table 3. Best until max. itr.: We run our solver for at most 10000 iterations and report best results (so $R = 50$, $T = 200$).

| instance | method | Until termination criteria | | | | Best until max. num itr. | | | |
|---|---|---|---|---|---|---|---|---|---|
| | | $E (\uparrow)$ | $g(t) (\downarrow)$ | $t (\downarrow)$ | # itr. | $E (\uparrow)$ | $g(t) (\downarrow)$ | $t (\downarrow)$ | # itr. |
| worm10-16-03-11-1745 | Gurobi | - | - | - | - | $-42557$ | 0 | 2356 | - |
| | DOGE | $-42645$ | 0.00305 | 35 | 5200 | $-42631$ | 0.0026 | 65 | 9600 |
| | DOGE-M | $-42611$ | 0.00188 | 37.5 | 5600 | $-42599$ | 0.00148 | 60 | 8800 |
| worm11-16-03-11-1745 | Gurobi | - | - | - | - | $-48672$ | 0 | 220 | - |
| | DOGE | $-48677$ | 0.00015 | 12.5 | 3000 | $-48675$ | 0.0001 | 25 | 5800 |
| | DOGE-M | $-48674$ | 0.00006 | 27.5 | 6400 | $-48674$ | 0.00005 | 42.5 | 9800 |
| worm12-16-03-11-1745 | Gurobi | - | - | - | - | $-50411$ | 0 | 68 | - |
| | DOGE | $-50411$ | 0 | 22.5 | 4800 | $-50411$ | 0 | 22.5 | 4800 |
| | DOGE-M | $-50411$ | 0 | 25 | 5400 | $-50411$ | 0 | 25 | 5400 |
| worm13-16-03-11-1745 | Gurobi | - | - | - | - | $-45836$ | 0 | 265 | - |
| | DOGE | $-45837$ | 0.00003 | 15 | 3800 | $-45837$ | 0.00003 | 15 | 3800 |
| | DOGE-M | $-45836$ | 0 | 17.5 | 4600 | $-45836$ | 0 | 27.5 | 7200 |
| worm14-16-03-11-1745 | Gurobi | - | - | - | - | $-47092$ | 0 | 509 | - |
| | DOGE | $-47108$ | 0.00058 | 20 | 4400 | $-47100$ | 0.00029 | 27.5 | 6000 |
| | DOGE-M | $-47100$ | 0.00027 | 42.5 | 9400 | $-47100$ | 0.00027 | 42.5 | 9400 |
| worm15-16-03-11-1745 | Gurobi | - | - | - | - | $-49551$ | 0 | 63 | - |
| | DOGE | $-49551$ | 0 | 12.5 | 3200 | $-49551$ | 0 | 12.5 | 3200 |
| | DOGE-M | $-49551$ | 0 | 12.5 | 3200 | $-49551$ | 0 | 12.5 | 3200 |
| worm16-16-03-11-1745 | Gurobi | - | - | - | - | $-48423$ | 0 | 238 | - |
| | DOGE | $-48428$ | 0.00019 | 15 | 3800 | $-48427$ | 0.00014 | 30 | 7400 |
| | DOGE-M | $-48425$ | 0.00009 | 15 | 3800 | $-48424$ | 0.00004 | 22.5 | 5600 |
| worm17-16-03-11-1745 | Gurobi | - | - | - | - | $-48082$ | 0 | 118 | - |
| | DOGE | $-48083$ | 0.00001 | 17.5 | 4200 | $-48082$ | 0 | 37.5 | 8800 |
| | DOGE-M | $-48083$ | 0.00003 | 12.5 | 2800 | $-48082$ | 0 | 20 | 4600 |
| worm18-16-03-11-1745 | Gurobi | - | - | - | - | $-48242$ | 0 | 98 | - |
| | DOGE | $-48242$ | 0.00001 | 25 | 5200 | $-48242$ | 0 | 32.5 | 6800 |
| | DOGE-M | $-48242$ | 0 | 12.5 | 2600 | $-48242$ | 0 | 12.5 | 2600 |
| worm19-16-03-11-1745 | Gurobi | - | - | - | - | $-48804$ | 0 | 195 | - |
| | DOGE | $-48807$ | 0.00011 | 15 | 3400 | $-48806$ | 0.00008 | 32.5 | 7200 |
| | DOGE-M | $-48806$ | 0.00008 | 17.5 | 3800 | $-48805$ | 0.00004 | 42.5 | 9400 |
| worm20-16-03-11-1745 | Gurobi | - | - | - | - | $-49443$ | 0 | 216 | - |
| | DOGE | $-49445$ | 0.00009 | 15 | 3000 | $-49443$ | 0.00001 | 42.5 | 8800 |
| | DOGE-M | $-49444$ | 0.00006 | 37.5 | 7800 | $-49444$ | 0.00006 | 37.5 | 7800 |
| worm21-16-03-11-1745 | Gurobi | - | - | - | - | $-49844$ | 0 | 67 | - |
| | DOGE | $-49844$ | 0 | 20 | 4400 | $-49844$ | 0 | 20 | 4400 |
| | DOGE-M | $-49844$ | 0 | 20 | 4400 | $-49844$ | 0 | 20 | 4400 |
| worm22-16-03-11-1745 | Gurobi | - | - | - | - | $-48012$ | 0 | 277 | - |
| | DOGE | $-48018$ | 0.00022 | 17.5 | 4200 | $-48013$ | 0.00002 | 40 | 9600 |
| | DOGE-M | $-48014$ | 0.00009 | 20 | 4800 | $-48013$ | 0.00003 | 32.5 | 7800 |
| worm23-16-03-11-1745 | Gurobi | - | - | - | - | $-49986$ | 0 | 51 | - |
| | DOGE | $-49986$ | 0 | 10 | 2200 | $-49986$ | 0 | 10 | 2200 |
| | DOGE-M | $-49986$ | 0 | 7.5 | 1600 | $-49986$ | 0 | 7.5 | 1600 |
| | Gurobi | - | - | - | - | $-49330$ | 0 | 79 | - |

*Continued on next page*

| instance | method | Until termination criteria | | | | Best until max. num itr. | | | |
|---|---|---|---|---|---|---|---|---|---|
| | | $E$ (↑) | $g(t)$ (↓) | $t$ (↓) | # itr. | $E$ (↑) | $g(t)$ (↓) | $t$ (↓) | # itr. |
| worm24-16-03-11-1745 | DOGE | −49333 | 0.00012 | 22.5 | 4800 | −49333 | 0.00012 | 22.5 | 4800 |
| | DOGE-M | −49330 | 0.00002 | 27.5 | 6000 | −49330 | 0.00001 | 37.5 | 8000 |
| | Gurobi | - | - | - | - | −47241 | 0 | 205 | - |
| worm25-16-03-11-1745 | DOGE | −47242 | 0.00002 | 17.5 | 4200 | −47241 | 0 | 30 | 7200 |
| | DOGE-M | −47242 | 0.00002 | 27.5 | 6600 | −47241 | 0 | 37.5 | 9200 |
| | Gurobi | - | - | - | - | −46145 | 0 | 595 | - |
| worm26-16-03-11-1745 | DOGE | −46161 | 0.00055 | 17.5 | 4000 | −46158 | 0.00046 | 35 | 8000 |
| | DOGE-M | −46150 | 0.00019 | 30 | 6800 | −46148 | 0.00011 | 42.5 | 9600 |
| | Gurobi | - | - | - | - | −50063 | 0 | 60 | - |
| worm27-16-03-11-1745 | DOGE | −50063 | 0 | 12.5 | 2600 | −50063 | 0 | 12.5 | 2600 |
| | DOGE-M | −50063 | 0 | 12.5 | 2600 | −50063 | 0 | 12.5 | 2600 |
| | Gurobi | - | - | - | - | −49500 | 0 | 59 | - |
| worm28-16-03-11-1745 | DOGE | −49500 | 0.00002 | 15 | 3400 | −49500 | 0.00002 | 25 | 5600 |
| | DOGE-M | −49500 | 0.00001 | 15 | 3200 | −49500 | 0 | 27.5 | 6000 |
| | Gurobi | - | - | - | - | −50070 | 0 | 46 | - |
| worm29-16-03-11-1745 | DOGE | −50070 | 0 | 15 | 3000 | −50070 | 0 | 15 | 3000 |
| | DOGE-M | −50070 | 0.00001 | 17.5 | 3400 | −50070 | 0 | 27.5 | 5400 |
| | Gurobi | - | - | - | - | −49784 | 0 | 58 | - |
| worm30-16-03-11-1745 | DOGE | −49784 | 0 | 12.5 | 2800 | −49784 | 0 | 12.5 | 2800 |
| | DOGE-M | −49784 | 0 | 15 | 3400 | −49784 | 0 | 15 | 3400 |

576 ## D.4  *QAPLib*

Table 7: Detailed results on *QAPLib* dataset. Until termination criteria contain results where we stop our solvers early w.r.t. relative improvement. These results are averaged and reported in Table 3. Best until max. itr.: We run our solver for at most 100000 iterations and report best results (so $R = 5000$, $T = 20$). *: Gurobi did not converge within 1 hour timelimit.

| instance | method | Until termination criteria | | | | Best until max. num itr. | | | |
|---|---|---|---|---|---|---|---|---|---|
| | | $E$ (↑) | $g(t)$ (↓) | $t$ (↓) | # itr. | $E$ (↑) | $g(t)$ (↓) | $t$ (↓) | # itr. |
| | Gurobi | - | - | - | - | 9886478 | 0.01997 | 3599 | - |
| bur26g* | DOGE | 10018869 | 0.00566 | 45 | 4820 | 10054780 | 0.00177 | 935 | 100000 |
| | DOGE-M | 10010474 | 0.00656 | 170 | 46340 | 10014676 | 0.00611 | 235 | 64080 |
| | Gurobi | - | - | - | - | 6060753 | 0.1538 | 3600 | - |
| bur26h* | DOGE | 7005771 | 0.008 | 50 | 5360 | 7034310 | 0.0036 | 930 | 99740 |
| | DOGE-M | 6997285 | 0.00931 | 185 | 50540 | 7001718 | 0.00863 | 250 | 68320 |
| | Gurobi | - | - | - | - | 6402 | 0.02938 | 3600 | - |
| had20* | DOGE | 6495 | 0.01392 | 280 | 62680 | 6512 | 0.0112 | 450 | 100000 |
| | DOGE-M | 6487 | 0.01532 | 225 | 99220 | 6487 | 0.01522 | 230 | 100000 |
| | Gurobi | - | - | - | - | 7703 | 0 | 3333 | - |
| kra32 | DOGE | 7481 | 0.0317 | 115 | 12260 | 7545 | 0.02259 | 940 | 100000 |
| | DOGE-M | 7457 | 0.03509 | 360 | 100000 | 7457 | 0.03509 | 360 | 100000 |
| | Gurobi | - | - | - | - | 4217 | 0.91943 | 3597 | - |
| lipa40a* | DOGE | 31506 | 0.00109 | 255 | 5480 | 31538 | 0 | 2465 | 52920 |
| | DOGE-M | 31417 | 0.00406 | 175 | 15300 | 31537 | 0.00004 | 1145 | 100000 |
| | Gurobi | - | - | - | - | 46637 | 0.92619 | 3598 | - |

*Continued on next page*

| instance | method | Until termination criteria | | | | Best until max. num itr. | | | |
|---|---|---|---|---|---|---|---|---|---|
| | | $E$ ($\uparrow$) | $g(t)$ ($\downarrow$) | $t$ ($\downarrow$) | # itr. | $E$ ($\uparrow$) | $g(t)$ ($\downarrow$) | $t$ ($\downarrow$) | # itr. |
| lipa40b* | DOGE | 439236 | 0.08045 | 1140 | 24980 | 442771 | 0.07284 | 1495 | 32760 |
| | DOGE-M | 471432 | 0.01109 | 485 | 43340 | 474399 | 0.0047 | 1120 | 100000 |
| | Gurobi | - | - | - | - | 3494 | 0.98823 | 3598 | - |
| lipa50a* | DOGE | 58664 | 0.05782 | 905 | 38720 | 60010 | 0.03512 | 2340 | 100000 |
| | DOGE-M | 61497 | 0.01005 | 145 | 5840 | 62093 | 0 | 1665 | 66480 |
| | Gurobi | - | - | - | - | 51648 | 0.97176 | 3595 | - |
| lipa50b* | DOGE | 1070018 | 0.10392 | 2310 | 99900 | 1070103 | 0.10385 | 2315 | 100000 |
| | DOGE-M | 1173647 | 0.01561 | 1190 | 47280 | 1191963 | 0 | 2490 | 100000 |
| | Gurobi | - | - | - | - | 3713 | 1 | 3595 | - |
| lipa60a* | DOGE | 105267 | 0.01789 | 735 | 15800 | 106426 | 0.00667 | 4660 | 100000 |
| | DOGE-M | 105786 | 0.01287 | 315 | 6420 | 107114 | 0 | 4955 | 100000 |
| | Gurobi | - | - | - | - | 66471 | 0.98356 | 3596 | - |
| lipa60b* | DOGE | 2093148 | 0.15489 | 145 | 3180 | 2186837 | 0.11658 | 4585 | 100000 |
| | DOGE-M | 2328269 | 0.05875 | 520 | 10720 | 2471953 | 0 | 4860 | 100000 |
| | Gurobi | - | - | - | - | 6598 | 0.99197 | 3600 | - |
| lipa70a* | DOGE | 165123 | 0.02784 | 1140 | 13160 | 167966 | 0.01054 | 8625 | 100000 |
| | DOGE-M | 167322 | 0.01446 | 565 | 6000 | 169700 | 0 | 9495 | 100000 |
| | Gurobi | - | - | - | - | 121986 | 0.98152 | 3600 | - |
| lipa70b* | DOGE | 4293967 | 0.03577 | 1990 | 23520 | 4382764 | 0.01564 | 4625 | 55120 |
| | DOGE-M | 4230582 | 0.05014 | 1355 | 14620 | 4451768 | 0 | 9310 | 100000 |
| | Gurobi | - | - | - | - | 2545 | 0.10356 | 3599 | - |
| nug27* | DOGE | 2693 | 0.04496 | 425 | 50200 | 2713 | 0.03731 | 850 | 100000 |
| | DOGE-M | 2688 | 0.04713 | 335 | 100000 | 2688 | 0.04713 | 335 | 100000 |
| | Gurobi | - | - | - | - | 2446 | 0.04303 | 3599 | - |
| nug28* | DOGE | 2468 | 0.03344 | 225 | 23300 | 2486 | 0.02522 | 965 | 100000 |
| | DOGE-M | 2456 | 0.03891 | 360 | 99580 | 2456 | 0.03884 | 365 | 100000 |
| | Gurobi | - | - | - | - | 1595 | 0.70017 | 3600 | - |
| nug30* | DOGE | 4481 | 0.07076 | 760 | 48520 | 4535 | 0.0588 | 1565 | 100000 |
| | DOGE-M | 4630 | 0.03817 | 510 | 99820 | 4630 | 0.03815 | 515 | 100000 |
| | Gurobi | - | - | - | - | 586646 | 0.09057 | 3599 | - |
| rou20* | DOGE | 612538 | 0.04922 | 445 | 99660 | 612560 | 0.04918 | 450 | 100000 |
| | DOGE-M | 612818 | 0.04877 | 225 | 99760 | 612844 | 0.04873 | 230 | 100000 |
| | Gurobi | - | - | - | - | 75474 | 0 | 43 | - |
| scr20 | DOGE | 75401 | 0.00132 | 45 | 15620 | 75415 | 0.00107 | 140 | 48680 |
| | DOGE-M | 75404 | 0.00126 | 30 | 16620 | 75415 | 0.00106 | 65 | 36600 |
| | Gurobi | - | - | - | - | 1599 | 0.8898 | 3599 | - |
| sko42* | DOGE | 9949 | 0.15128 | 1355 | 99980 | 9949 | 0.15125 | 1360 | 100000 |
| | DOGE-M | 11597 | 0.00557 | 1165 | 78220 | 11659 | 0 | 1480 | 100000 |
| | Gurobi | - | - | - | - | 1268 | 0.94759 | 3599 | - |
| sko49* | DOGE | 15745 | 0.05392 | 2210 | 99800 | 15747 | 0.05383 | 2215 | 100000 |
| | DOGE-M | 16439 | 0.01109 | 1650 | 70580 | 16619 | 0 | 2340 | 100000 |
| | Gurobi | - | - | - | - | 1421 | 0.96053 | 3594 | - |
| sko56* | DOGE | 22410 | 0.06144 | 125 | 3380 | 23430 | 0.01774 | 3695 | 100000 |
| | DOGE-M | 23254 | 0.02529 | 2020 | 52180 | 23845 | 0 | 3875 | 100000 |
| | Gurobi | - | - | - | - | 1053 | 0.98536 | 3586 | - |
| sko64* | DOGE | 30410 | 0.0819 | 15 | 240 | 31798 | 0.03917 | 5915 | 99980 |

*Continued on next page*

| instance | method | Until termination criteria | | | | Best until max. num itr. | | | |
|---|---|---|---|---|---|---|---|---|---|
| | | $E$ ($\uparrow$) | $g(t)$ ($\downarrow$) | $t$ ($\downarrow$) | # itr. | $E$ ($\uparrow$) | $g(t)$ ($\downarrow$) | $t$ ($\downarrow$) | # itr. |
| | DOGE-M | 31517 | 0.04784 | 2065 | 33000 | 33071 | 0 | 6255 | 100000 |
| ste36c* | Gurobi | - | - | - | - | 2661689 | 0.65051 | 3599 | - |
| | DOGE | 6759633 | 0.0412 | 820 | 26300 | 6967435 | 0.0103 | 3110 | 99660 |
| | DOGE-M | 6933566 | 0.01533 | 485 | 60140 | 6976984 | 0.00888 | 765 | 94840 |
| tai35a* | Gurobi | - | - | - | - | 180704 | 0.91506 | 3598 | - |
| | DOGE | 1794138 | 0.09976 | 1285 | 46320 | 1802223 | 0.09568 | 1405 | 50640 |
| | DOGE-M | 1896341 | 0.04812 | 735 | 99900 | 1896382 | 0.0481 | 740 | 100000 |
| tai35b* | Gurobi | - | - | - | - | 5212148 | 0.9588 | 3601 | - |
| | DOGE | 96397776 | 0.07721 | 540 | 19280 | 99942776 | 0.04294 | 2795 | 99880 |
| | DOGE-M | 98365992 | 0.05818 | 750 | 100000 | 98365992 | 0.05818 | 750 | 100000 |
| tai40a* | Gurobi | - | - | - | - | 121132 | 0.95377 | 3599 | - |
| | DOGE | 2140900 | 0.07833 | 480 | 42920 | 2142402 | 0.07768 | 595 | 53280 |
| | DOGE-M | 2321628 | 0 | 1165 | 100000 | 2321628 | 0 | 1165 | 100000 |
| tai40b* | Gurobi | - | - | - | - | 5241527 | 0.97122 | 3599 | - |
| | DOGE | 121210376 | 0.09553 | 295 | 25960 | 121210376 | 0.09553 | 295 | 25960 |
| | DOGE-M | 133862088 | 0 | 1185 | 100000 | 133862088 | 0 | 1185 | 100000 |
| tai50a* | Gurobi | - | - | - | - | 61532 | 0.98721 | 3592 | - |
| | DOGE | 2988845 | 0.18113 | 535 | 23040 | 2994846 | 0.17948 | 1610 | 69440 |
| | DOGE-M | 3622177 | 0.00673 | 2125 | 87480 | 3646619 | 0 | 2435 | 100000 |
| tai50b* | Gurobi | - | - | - | - | 179580 | 1 | 3599 | - |
| | DOGE | 83312440 | 0.1107 | 1995 | 84120 | 85635848 | 0.08563 | 2375 | 100000 |
| | DOGE-M | 93571496 | 0 | 2480 | 100000 | 93571496 | 0 | 2480 | 100000 |
| tai60a* | Gurobi | - | - | - | - | 87106 | 0.98697 | 3593 | - |
| | DOGE | 3984766 | 0.26193 | 265 | 5620 | 4319060 | 0.19975 | 4625 | 100000 |
| | DOGE-M | 5216043 | 0.03289 | 1840 | 38040 | 5392868 | 0 | 4860 | 100000 |
| tai60b* | Gurobi | - | - | - | - | 125579 | 1 | 3590 | - |
| | DOGE | 55250419 | 0.60577 | 255 | 5520 | 88722336 | 0.36445 | 4625 | 99960 |
| | DOGE-M | 106619936 | 0.23542 | 915 | 18680 | 139274272 | 0 | 4910 | 100000 |
| tai64c | Gurobi | - | - | - | - | 487500 | 0 | 3283 | - |
| | DOGE | 482685 | 0.01197 | 5 | 800 | 487483 | 0.00004 | 275 | 43760 |
| | DOGE-M | 486733 | 0.00191 | 25 | 760 | 486733 | 0.00191 | 25 | 760 |
| tho30* | Gurobi | - | - | - | - | 33467 | 0.70179 | 3598 | - |
| | DOGE | 90078 | 0.11192 | 945 | 60560 | 91072 | 0.10157 | 1560 | 100000 |
| | DOGE-M | 95420 | 0.05626 | 510 | 100000 | 95420 | 0.05626 | 510 | 100000 |
| wil50* | Gurobi | - | - | - | - | 3037 | 0.94051 | 3597 | - |
| | DOGE | 35943 | 0.08066 | 1655 | 24640 | 36941 | 0.05458 | 6715 | 100000 |
| | DOGE-M | 38775 | 0.00667 | 2140 | 85100 | 39030 | 0 | 2515 | 100000 |

577 **D.5** *Independent set*

Table 8: Detailed results on *Independent set* dataset. Until termination criteria contain results where we stop our solvers early w.r.t. relative improvement. These results are averaged and reported in Table 3. Best until max. itr.: We run our solver for at most 10000 iterations and report best results (so $R = 200$, $T = 50$).

| instance | method | Until termination criteria | | | | Best until max. num itr. | | | |
|---|---|---|---|---|---|---|---|---|---|
| | | $E$ (↑) | $g(t)$ (↓) | $t$ (↓) | # itr. | $E$ (↑) | $g(t)$ (↓) | $t$ (↓) | # itr. |
| 1 | Gurobi | - | - | - | - | $-24444$ | 0 | 50 | - |
| | DOGE | $-24447$ | 0.0057 | 4.2 | 1550 | $-24445$ | 0.00243 | 10.1 | 3800 |
| | DOGE-M | $-24445$ | 0.00244 | 2.3 | 850 | $-24444$ | 0.00152 | 3.5 | 1300 |
| 10 | Gurobi | - | - | - | - | $-24457$ | 0 | 47 | - |
| | DOGE | $-24465$ | 0.01482 | 4 | 1450 | $-24459$ | 0.00427 | 7.3 | 2650 |
| | DOGE-M | $-24476$ | 0.03372 | 1 | 350 | $-24459$ | 0.00433 | 2.2 | 800 |
| 11 | Gurobi | - | - | - | - | $-24464$ | 0 | 48 | - |
| | DOGE | $-24468$ | 0.0075 | 4.2 | 1550 | $-24465$ | 0.00276 | 11.1 | 4100 |
| | DOGE-M | $-24465$ | 0.0032 | 3.2 | 1200 | $-24465$ | 0.0023 | 4.4 | 1650 |
| 12 | Gurobi | - | - | - | - | $-24453$ | 0 | 56 | - |
| | DOGE | $-24460$ | 0.01384 | 2.7 | 1000 | $-24454$ | 0.00268 | 14 | 5200 |
| | DOGE-M | $-24476$ | 0.04099 | 1.1 | 400 | $-24454$ | 0.00237 | 9.5 | 3350 |
| 13 | Gurobi | - | - | - | - | $-24461$ | 0 | 56 | - |
| | DOGE | $-24466$ | 0.00895 | 4 | 1500 | $-24463$ | 0.00363 | 11.5 | 4300 |
| | DOGE-M | $-24484$ | 0.04179 | 1.1 | 400 | $-24462$ | 0.00202 | 5.3 | 2000 |
| 14 | Gurobi | - | - | - | - | $-24455$ | 0 | 53 | - |
| | DOGE | $-24461$ | 0.0106 | 3.2 | 1150 | $-24456$ | 0.00177 | 13.4 | 4900 |
| | DOGE-M | $-24458$ | 0.00629 | 2.5 | 900 | $-24458$ | 0.00494 | 10.3 | 3800 |
| 15 | Gurobi | - | - | - | - | $-24467$ | 0 | 50 | - |
| | DOGE | $-24474$ | 0.01292 | 3.3 | 1200 | $-24469$ | 0.00293 | 10.8 | 4000 |
| | DOGE-M | $-24469$ | 0.00397 | 2.6 | 950 | $-24469$ | 0.00277 | 6.8 | 2500 |
| 16 | Gurobi | - | - | - | - | $-24452$ | 0 | 50 | - |
| | DOGE | $-24457$ | 0.00917 | 3 | 1100 | $-24454$ | 0.00437 | 10.2 | 3750 |
| | DOGE-M | $-24453$ | 0.00259 | 2.3 | 850 | $-24452$ | 0.00102 | 6 | 2250 |
| 17 | Gurobi | - | - | - | - | $-24452$ | 0 | 50 | - |
| | DOGE | $-24457$ | 0.00963 | 4.5 | 1650 | $-24454$ | 0.00401 | 14.8 | 5500 |
| | DOGE-M | $-24454$ | 0.00342 | 2.6 | 950 | $-24453$ | 0.00212 | 7.9 | 2900 |
| 18 | Gurobi | - | - | - | - | $-24473$ | 0 | 45 | - |
| | DOGE | $-24487$ | 0.02501 | 2.8 | 1000 | $-24476$ | 0.00555 | 11.3 | 4050 |
| | DOGE-M | $-24475$ | 0.00447 | 2.7 | 950 | $-24474$ | 0.00248 | 7.7 | 2800 |
| 19 | Gurobi | - | - | - | - | $-24458$ | 0 | 59 | - |
| | DOGE | $-24467$ | 0.01653 | 2.6 | 950 | $-24460$ | 0.00368 | 12.6 | 4650 |
| | DOGE-M | $-24477$ | 0.03407 | 1.1 | 400 | $-24459$ | 0.00211 | 3 | 1100 |
| 2 | Gurobi | - | - | - | - | $-24459$ | 0 | 45 | - |
| | DOGE | $-24464$ | 0.00978 | 3.5 | 1250 | $-24460$ | 0.00331 | 14.4 | 5350 |
| | DOGE-M | $-24464$ | 0.0102 | 2.3 | 850 | $-24464$ | 0.0102 | 2.3 | 850 |
| 20 | Gurobi | - | - | - | - | $-24458$ | 0 | 55 | - |
| | DOGE | $-24466$ | 0.01333 | 2.6 | 950 | $-24460$ | 0.00267 | 12.5 | 4650 |
| | DOGE-M | $-24460$ | 0.00316 | 2.4 | 900 | $-24459$ | 0.0026 | 3.7 | 1350 |
| 21 | Gurobi | - | - | - | - | $-24458$ | 0 | 55 | - |
| | DOGE | $-24470$ | 0.02143 | 4.8 | 850 | $-24459$ | 0.0033 | 14.9 | 4550 |
| | DOGE-M | $-24459$ | 0.00269 | 2.4 | 850 | $-24458$ | 0.00137 | 3.4 | 1200 |
| | Gurobi | - | - | - | - | $-24460$ | 0 | 64 | - |

*Continued on next page*

| instance | method | Until termination criteria | | | | Best until max. num itr. | | | |
|---|---|---|---|---|---|---|---|---|---|
| | | $E$ (↑) | $g(t)$ (↓) | $t$ (↓) | # itr. | $E$ (↑) | $g(t)$ (↓) | $t$ (↓) | # itr. |
| 22 | DOGE | −24465 | 0.00884 | 3.7 | 1350 | −24462 | 0.00414 | 12.5 | 4600 |
| | DOGE-M | −24462 | 0.00313 | 3.6 | 1350 | −24462 | 0.00286 | 6.1 | 2250 |
| | Gurobi | - | - | - | - | −24438 | 0 | 51 | - |
| 23 | DOGE | −24441 | 0.00678 | 4.8 | 1800 | −24440 | 0.00385 | 8.9 | 3300 |
| | DOGE-M | −24441 | 0.00617 | 2.3 | 850 | −24439 | 0.00307 | 11.3 | 4150 |
| | Gurobi | - | - | - | - | −24437 | 0 | 52 | - |
| 24 | DOGE | −24448 | 0.01857 | 2.4 | 850 | −24439 | 0.00274 | 10.9 | 4000 |
| | DOGE-M | −24455 | 0.03095 | 1.1 | 400 | −24439 | 0.00296 | 3.7 | 1350 |
| | Gurobi | - | - | - | - | −24468 | 0 | 47 | - |
| 25 | DOGE | −24477 | 0.0178 | 2.7 | 1000 | −24470 | 0.00371 | 10.8 | 4000 |
| | DOGE-M | −24489 | 0.03955 | 1.1 | 400 | −24469 | 0.00317 | 15.4 | 5700 |
| | Gurobi | - | - | - | - | −24444 | 0 | 56 | - |
| 26 | DOGE | −24449 | 0.00899 | 3.4 | 1250 | −24446 | 0.004 | 10.3 | 3850 |
| | DOGE-M | −24447 | 0.00442 | 1.9 | 700 | −24446 | 0.00346 | 3.9 | 1450 |
| | Gurobi | - | - | - | - | −24466 | 0 | 53 | - |
| 27 | DOGE | −24472 | 0.0107 | 4.1 | 1500 | −24469 | 0.00523 | 15 | 5500 |
| | DOGE-M | −24468 | 0.00308 | 2.1 | 750 | −24467 | 0.00128 | 4.6 | 1700 |
| | Gurobi | - | - | - | - | −24472 | 0 | 51 | - |
| 28 | DOGE | −24477 | 0.00927 | 4 | 1300 | −24473 | 0.00352 | 12.2 | 4300 |
| | DOGE-M | −24492 | 0.03674 | 1 | 350 | −24473 | 0.00265 | 6 | 2200 |
| | Gurobi | - | - | - | - | −24446 | 0 | 52 | - |
| 29 | DOGE | −24453 | 0.01268 | 3.3 | 1200 | −24448 | 0.00365 | 11.1 | 4050 |
| | DOGE-M | −24447 | 0.00231 | 2.6 | 950 | −24446 | 0.00056 | 4.2 | 1500 |
| | Gurobi | - | - | - | - | −24457 | 0 | 50 | - |
| 3 | DOGE | −24465 | 0.01357 | 3 | 1100 | −24459 | 0.00273 | 10.6 | 3900 |
| | DOGE-M | −24479 | 0.03901 | 1.4 | 450 | −24458 | 0.0022 | 4.6 | 1550 |
| | Gurobi | - | - | - | - | −24454 | 0 | 63 | - |
| 30 | DOGE | −24458 | 0.00777 | 5.3 | 1950 | −24456 | 0.00296 | 13.7 | 5050 |
| | DOGE-M | −24483 | 0.0518 | 1.4 | 500 | −24456 | 0.00309 | 10.8 | 3950 |
| | Gurobi | - | - | - | - | −24457 | 0 | 47 | - |
| 31 | DOGE | −24465 | 0.01361 | 3.5 | 1250 | −24459 | 0.00414 | 10.9 | 3950 |
| | DOGE-M | −24459 | 0.00294 | 3 | 1100 | −24458 | 0.0019 | 7.2 | 2650 |
| | Gurobi | - | - | - | - | −24466 | 0 | 41 | - |
| 32 | DOGE | −24474 | 0.01457 | 2.9 | 1050 | −24468 | 0.00311 | 11.2 | 4100 |
| | DOGE-M | −24487 | 0.03785 | 1.3 | 400 | −24468 | 0.00273 | 8.4 | 3000 |
| | Gurobi | - | - | - | - | −24452 | 0 | 70 | - |
| 4 | DOGE | −24457 | 0.00847 | 4.4 | 1550 | −24454 | 0.00316 | 13.3 | 4800 |
| | DOGE-M | −24454 | 0.0041 | 1.8 | 650 | −24454 | 0.00403 | 2.3 | 850 |
| | Gurobi | - | - | - | - | −24474 | 0 | 49 | - |
| 5 | DOGE | −24482 | 0.01549 | 2.3 | 850 | −24476 | 0.003 | 11.1 | 4100 |
| | DOGE-M | −24494 | 0.03738 | 1.1 | 400 | −24475 | 0.00205 | 4.5 | 1650 |
| | Gurobi | - | - | - | - | −24474 | 0 | 49 | - |
| 59 | DOGE | −24478 | 0.0074 | 4.1 | 1500 | −24476 | 0.00298 | 9.9 | 3650 |
| | DOGE-M | −24494 | 0.03699 | 1.1 | 400 | −24475 | 0.00184 | 7.1 | 2600 |
| | Gurobi | - | - | - | - | −24468 | 0 | 52 | - |
| 6 | DOGE | −24472 | 0.00858 | 4.5 | 1500 | −24469 | 0.00273 | 10.8 | 3850 |

*Continued on next page*

| instance | method | Until termination criteria | | | | Best until max. num itr. | | | |
|---|---|---|---|---|---|---|---|---|---|
| | | $E$ ($\uparrow$) | $g(t)$ ($\downarrow$) | $t$ ($\downarrow$) | # itr. | $E$ ($\uparrow$) | $g(t)$ ($\downarrow$) | $t$ ($\downarrow$) | # itr. |
| | DOGE-M | −24468 | 0.00156 | 2.9 | 1000 | −24468 | 0.00068 | 6.2 | 2150 |
| 60 | Gurobi | - | - | - | - | −24468 | 0 | 52 | - |
| | DOGE | −24473 | 0.00936 | 3.5 | 1300 | −24469 | 0.00351 | 10.8 | 4100 |
| | DOGE-M | −24469 | 0.00186 | 1.9 | 700 | −24468 | 0.00024 | 3.7 | 1400 |
| 61 | Gurobi | - | - | - | - | −24470 | 0 | 49 | - |
| | DOGE | −24474 | 0.00867 | 4 | 1500 | −24471 | 0.00237 | 13.8 | 5100 |
| | DOGE-M | −24472 | 0.00363 | 3.1 | 1150 | −24471 | 0.0029 | 6 | 2200 |
| 62 | Gurobi | - | - | - | - | −24456 | 0 | 48 | - |
| | DOGE | −24471 | 0.02682 | 2.5 | 900 | −24459 | 0.00569 | 13.5 | 4900 |
| | DOGE-M | −24461 | 0.00914 | 2.8 | 1000 | −24459 | 0.00585 | 13 | 4750 |
| 63 | Gurobi | - | - | - | - | −24442 | 0 | 44 | - |
| | DOGE | −24447 | 0.00968 | 3.9 | 1450 | −24444 | 0.00374 | 11.4 | 4150 |
| | DOGE-M | −24443 | 0.00289 | 3.1 | 1150 | −24443 | 0.00235 | 9.6 | 3500 |
| 64 | Gurobi | - | - | - | - | −24457 | 0 | 48 | - |
| | DOGE | −24466 | 0.0173 | 3.3 | 1200 | −24458 | 0.00313 | 11.3 | 4150 |
| | DOGE-M | −24476 | 0.03412 | 1.1 | 400 | −24459 | 0.00416 | 2.9 | 1050 |
| 65 | Gurobi | - | - | - | - | −24464 | 0 | 48 | - |
| | DOGE | −24468 | 0.00831 | 4.5 | 1650 | −24465 | 0.00299 | 10.9 | 4050 |
| | DOGE-M | −24489 | 0.04566 | 1 | 350 | −24465 | 0.00211 | 8.1 | 3000 |
| 66 | Gurobi | - | - | - | - | −24453 | 0 | 55 | - |
| | DOGE | −24456 | 0.00653 | 4.3 | 1600 | −24454 | 0.00247 | 10.9 | 4050 |
| | DOGE-M | −24455 | 0.00379 | 3.5 | 1300 | −24454 | 0.00223 | 5.4 | 2000 |
| 67 | Gurobi | - | - | - | - | −24461 | 0 | 56 | - |
| | DOGE | −24469 | 0.01437 | 2.8 | 1050 | −24462 | 0.00308 | 14 | 5250 |
| | DOGE-M | −24484 | 0.042 | 1.1 | 400 | −24462 | 0.00176 | 6.3 | 2350 |
| 68 | Gurobi | - | - | - | - | −24455 | 0 | 54 | - |
| | DOGE | −24461 | 0.011 | 3 | 1100 | −24456 | 0.00235 | 13.2 | 4900 |
| | DOGE-M | −24458 | 0.00509 | 2.5 | 900 | −24458 | 0.00506 | 3.1 | 1150 |
| 69 | Gurobi | - | - | - | - | −24467 | 0 | 51 | - |
| | DOGE | −24477 | 0.01798 | 1.9 | 700 | −24468 | 0.00145 | 8.8 | 3250 |
| | DOGE-M | −24487 | 0.03673 | 1.1 | 400 | −24468 | 0.002 | 4.2 | 1550 |
| 7 | Gurobi | - | - | - | - | −24470 | 0 | 51 | - |
| | DOGE | −24476 | 0.01197 | 3.1 | 1150 | −24471 | 0.00282 | 11.8 | 4400 |
| | DOGE-M | −24489 | 0.03455 | 1.3 | 450 | −24471 | 0.00269 | 8.1 | 3000 |
| 70 | Gurobi | - | - | - | - | −24452 | 0 | 50 | - |
| | DOGE | −24459 | 0.0137 | 2.6 | 950 | −24454 | 0.0041 | 11.4 | 4200 |
| | DOGE-M | −24469 | 0.03031 | 1.1 | 400 | −24452 | 0.00104 | 7.6 | 2800 |
| 71 | Gurobi | - | - | - | - | −24452 | 0 | 48 | - |
| | DOGE | −24464 | 0.02105 | 2.2 | 800 | −24453 | 0.0023 | 10.5 | 3900 |
| | DOGE-M | −24468 | 0.028 | 1.5 | 550 | −24454 | 0.00272 | 8.4 | 3100 |
| 72 | Gurobi | - | - | - | - | −24473 | 0 | 45 | - |
| | DOGE | −24480 | 0.01316 | 3.7 | 1300 | −24477 | 0.00642 | 9.4 | 3350 |
| | DOGE-M | −24496 | 0.04223 | 1 | 350 | −24474 | 0.00207 | 5.3 | 1900 |
| 73 | Gurobi | - | - | - | - | −24458 | 0 | 57 | - |
| | DOGE | −24463 | 0.00993 | 3.3 | 1200 | −24460 | 0.00384 | 10.2 | 3750 |
| | DOGE-M | −24476 | 0.03355 | 1.1 | 400 | −24458 | 0.00158 | 5.1 | 1900 |

| instance | method | Until termination criteria | | | | Best until max. num itr. | | | |
|---|---|---|---|---|---|---|---|---|---|
| | | $E$ ($\uparrow$) | $g(t)$ ($\downarrow$) | $t$ ($\downarrow$) | # itr. | $E$ ($\uparrow$) | $g(t)$ ($\downarrow$) | $t$ ($\downarrow$) | # itr. |
| 74 | Gurobi | - | - | - | - | $-24458$ | 0 | 55 | - |
| | DOGE | $-24466$ | 0.01417 | 2.3 | 850 | $-24460$ | 0.00264 | 12.2 | 4550 |
| | DOGE-M | $-24459$ | 0.00177 | 2.5 | 900 | $-24459$ | 0.00172 | 5.5 | 1950 |
| 75 | Gurobi | - | - | - | - | $-24458$ | 0 | 55 | - |
| | DOGE | $-24464$ | 0.01114 | 3.7 | 1350 | $-24460$ | 0.00404 | 9.5 | 3450 |
| | DOGE-M | $-24460$ | 0.00437 | 2 | 650 | $-24459$ | 0.00226 | 3.7 | 1250 |
| 76 | Gurobi | - | - | - | - | $-24460$ | 0 | 63 | - |
| | DOGE | $-24465$ | 0.00857 | 3.7 | 1350 | $-24462$ | 0.0039 | 8.3 | 3050 |
| | DOGE-M | $-24462$ | 0.0036 | 3.3 | 1200 | $-24461$ | 0.00224 | 6 | 2200 |
| 77 | Gurobi | - | - | - | - | $-24438$ | 0 | 48 | - |
| | DOGE | $-24443$ | 0.00943 | 3.5 | 1300 | $-24440$ | 0.00379 | 13 | 4850 |
| | DOGE-M | $-24440$ | 0.00411 | 3 | 1100 | $-24439$ | 0.00293 | 9.4 | 3450 |
| 78 | Gurobi | - | - | - | - | $-24437$ | 0 | 52 | - |
| | DOGE | $-24446$ | 0.01493 | 3.4 | 1250 | $-24440$ | 0.00467 | 14.2 | 5250 |
| | DOGE-M | $-24439$ | 0.00286 | 2.7 | 1000 | $-24438$ | 0.00236 | 4.8 | 1800 |
| 79 | Gurobi | - | - | - | - | $-24468$ | 0 | 46 | - |
| | DOGE | $-24477$ | 0.01686 | 3.1 | 1150 | $-24470$ | 0.00453 | 11.9 | 4400 |
| | DOGE-M | $-24490$ | 0.04014 | 1.1 | 400 | $-24469$ | 0.00282 | 9.8 | 3600 |
| 8 | Gurobi | - | - | - | - | $-24456$ | 0 | 48 | - |
| | DOGE | $-24472$ | 0.02816 | 2.3 | 800 | $-24459$ | 0.00491 | 15.1 | 5450 |
| | DOGE-M | $-24460$ | 0.00756 | 2.9 | 1050 | $-24459$ | 0.0055 | 14.9 | 5450 |
| 80 | Gurobi | - | - | - | - | $-24444$ | 0 | 56 | - |
| | DOGE | $-24449$ | 0.00827 | 3.9 | 1450 | $-24446$ | 0.0042 | 11.2 | 4200 |
| | DOGE-M | $-24460$ | 0.02826 | 1.1 | 400 | $-24446$ | 0.00328 | 2.9 | 1100 |
| 81 | Gurobi | - | - | - | - | $-24466$ | 0 | 53 | - |
| | DOGE | $-24475$ | 0.01564 | 2.6 | 950 | $-24468$ | 0.00401 | 13.6 | 4950 |
| | DOGE-M | $-24467$ | 0.00184 | 2.9 | 1050 | $-24467$ | 0.001 | 4.3 | 1550 |
| 82 | Gurobi | - | - | - | - | $-24472$ | 0 | 51 | - |
| | DOGE | $-24481$ | 0.01659 | 2.2 | 800 | $-24473$ | 0.00265 | 12.3 | 4550 |
| | DOGE-M | $-24474$ | 0.00374 | 2 | 700 | $-24473$ | 0.00264 | 3.9 | 1450 |
| 83 | Gurobi | - | - | - | - | $-24446$ | 0 | 52 | - |
| | DOGE | $-24452$ | 0.01118 | 3.7 | 1350 | $-24447$ | 0.00281 | 11.8 | 4300 |
| | DOGE-M | $-24446$ | 0.00169 | 2.8 | 900 | $-24446$ | 0.00088 | 4.4 | 1450 |
| 84 | Gurobi | - | - | - | - | $-24454$ | 0 | 63 | - |
| | DOGE | $-24460$ | 0.01068 | 4.1 | 1500 | $-24456$ | 0.00339 | 11.2 | 4100 |
| | DOGE-M | $-24483$ | 0.0518 | 1.4 | 500 | $-24456$ | 0.00334 | 11.1 | 3850 |
| 85 | Gurobi | - | - | - | - | $-24457$ | 0 | 48 | - |
| | DOGE | $-24469$ | 0.02107 | 3.2 | 1150 | $-24460$ | 0.0051 | 10.4 | 3750 |
| | DOGE-M | $-24459$ | 0.00392 | 2.5 | 900 | $-24458$ | 0.00185 | 5.9 | 2150 |
| 86 | Gurobi | - | - | - | - | $-24466$ | 0 | 41 | - |
| | DOGE | $-24476$ | 0.01829 | 3.3 | 1200 | $-24469$ | 0.00534 | 9.5 | 3500 |
| | DOGE-M | $-24468$ | 0.00361 | 3.2 | 1150 | $-24468$ | 0.00281 | 4.4 | 1600 |
| 9 | Gurobi | - | - | - | - | $-24442$ | 0 | 44 | - |
| | DOGE | $-24448$ | 0.01164 | 3.4 | 1200 | $-24443$ | 0.0029 | 10.6 | 3850 |
| | DOGE-M | $-24443$ | 0.00229 | 3.3 | 1200 | $-24442$ | 0.00152 | 3.8 | 1350 |