# OpenReview forum: "DOGE-Train: Discrete Optimization on GPU with End-to-end Training"
_NeurIPS.cc/2022/Conference — NeurIPS 2022 Submitted_

### Official Review · Reviewer_jrD7 · 2022-07-10

**Rating:** 7
**Confidence:** 4
**Soundness:** 3 good
**Presentation:** 3 good
**Contribution:** 3 good

**Summary:**

This paper proposes a novel framework to speed up solving ILP problems through tuning parameters of Parallel Deferred Min-Marginal Averaging algorithm. This tuning is based on the gradients of the loss function w.r.t to these parameters, so the ILP solver is represented as a neural network that can be optimized with gradient-based optimization method. The los function here is explicit dual objective function. The parameters update preserves their feasibility and non-decrease of the lower-bound. Moreover, to prevent stuck in fixed point, non-parametric update step is also proposed and separate model to generate it is trained. The presented method is evaluated in four classes of combinatorial problems, shows faster convergence than Gurobi solver and establishes smaller relative duality gap.

**Questions:**

Please comment the following remarks and questions:
1) In line 34, authors claim that simplex or interior-point methods are not GPU friendly, however, they consist of series of linear algebra operations, that can be speed up due to GPU architectures e.g. matrix multiplication in dense layers of neural networks is faster on GPU rather than on CPU. Please, provide more detailed explanation of this point.
2) What is the motivation for forward and backward iterations over blocks $B_1,\ldots, B_u$ in algorithm 1?
3) Please provide more details about computing subgradients in eq 23 from Appendix. Since this is the single non-trivial operation in the differentiating of the considered method (others operations are just summation and subtraction), its detailed description is important for understanding the correctness of the proposed approach.
4) Alg 2 solves two optimization problems (for $\beta = 0$ and $\beta = 1$), so it is not clear the difference with the naive implementation. Please, provide more details or explicitly present the naive form and highlight the difference.
5) Please provide detailed description of datasets that are used in generating of Figure 2. Do these plots correspond to a picked instance of the corresponding problem class? Or some average over all considered instances are made?
6) What solver do authors use inside the proposed framework to solve problems in line 10, Alg 1 and in line 4, Alg 2?
7) Please explicitly describe how to transform standard constraints like $Ax \leq b$ to the form in problem (BP) ?
8) Please explicitly state and describe decreasing of what quantities can explain the observed speed up. Are they the overall number of iterations, acceleration of solving intermediate optimization problems with GPU, or anything else? Such comparison with other state-of-the-art methods helps to place the proposed framework in the proper context and improve understanding of the reasons for the observed gain. If such comparison is not possible for Gurobi since authors do not have access to its detailed statistics, it is possible for FastDog, which is non-differentiable version of the proposed approach with fixed parameters $\omega_{ij}$ and $\alpha_{ij}$.
9) I assume that in line 123 $\omega$ should be replaced with $\omega_{ij}$ according to the line 10 in Alg. 1

**Limitations:**

The authors explicitly state the problems where the proposed framework does not show improvement and discuss the possible reasons for that.

**Strengths And Weaknesses:**

Strengths:
1) extensive computational experiments for evaluation of the proposed method
2) the novel technique for differentiating optimization methods w.r.t. its parameters
3) recipes how to escape from the fixed points
4) unsupervised loss function eliminates any data preparation stage

Weaknesses:
1) hand-crafted features for GNN requires some approach to encode current state of Alg 1 in automatic way or some theory that explains the sufficiency of the presented set of features
2) it remains unclear why such complicated procedure that computes gradients, updates parameters of optimizer in two-ways (parametric and non-parametric) and requires additional memory to store parameters of neural networks is still faster than the state-of-the-art solvers like Gurobi.

---

> ### Author Response · Authors · 2022-07-29
> **Response to questions raised by reviewer jrD7**
>
>
> We address the raised questions as follows:
>
> ### 1. Simplex or interior-point methods are not GPU friendly?
> For the parallel simplex method we refer to the work of [HU18] (especially Sec. 6) where the authors were only able to obtain a $1.5x$ speed-up over open-source solvers and no speed-up over commercial solvers. For interior point method a major hurdle towards parallelization is the stage of matrix decomposition which is executed sequentially. Lastly and more importantly, for our work we require a differentiable solver but both interior-point and simplex method contain non-differentiable steps.
>
> [HU18] - Huangfu, Qi, and JA Julian Hall. "Parallelizing the dual revised simplex method." Mathematical Programming Computation 10.1 (2018): 119-142.
>
> ### 2. Motivation for forward and backward iterations in algorithm 1?
> These iterations mimic the variable update direction of FastDOG (line 8, Alg. 1). Intuitively the goal is to allow back and forth exchange among the dual variables. Note that each variable block can be run in parallel allowing efficient GPU implementation.
>
> ### 3. More details about computing subgradients in eq 23 from Appendix.
> The subgradient is computed by applying Danskin's theorem. We will add more details in the final version.
>
> ### 4. Alg 2 solves two optimization problems (for $\beta \in \{0, 1\}$).
> Actually in Alg. 1 and 2 we present a naive implementation agnostic way of computing min-marginals and their gradients.
> For implementation we actually use the efficient BDD-based scheme as given in Section 5.1 of FastDOG avoiding the computational overhead of solving optimization problems. In the BDD-based implementation the solutions of these two subproblems come at negligible additional cost.
>
> ### 5. Description of datasets that are used in generating of Figure 2.
> For each dataset the plot is generated by averaging over all of its test instances as also done in FastDOG.
>
> ### 6. What solver do authors use inside the proposed framework to solve problems in line 10, Alg 1 and in line 4, Alg 2?
> We use the efficient schemes for computing these terms following Section 5.1 in FastDOG. We will make it clear in the final version. Please also see our answer to question 4 above.
>
> ### 7. Please explicitly describe how to transform standard constraints like to the form in problem (BP)?
> We use the scheme of [31] which is also used by FastDOG. We will elaborate on it in the final manuscript.
>
> ### 8. Please explicitly state and describe decreasing of what quantities can explain the observed speed up.
> There are two points of comparisons of speed up which we breakdown as follows:
>
> 1. **Speed-up over FastDOG:** To perform a fixed number of iterations our implementation actually takes around $1.2x$ more time than FastDOG. This is mainly because we have a separate value of $\omega$ and $\alpha$ for each dual variable instead of only a scalar for FastDOG. Since the GNN can predict good values for these parameters and non-parametric update (eq. 3) we are able to achieve much large improvement in dual objective per iteration as compared to FastDOG. We will add iteration level comparison in the final version.
>
> 2. **Speed-up over Gurobi:** We build on FastDOG which performs parallel computation and already outperforms Gurobi at early stages of optimization. However, FastDOG can get stuck in suboptimal fixed points far away from the optimum. Our learned scheme allows to circumvent this issue allowing us to surpass Gurobi also at the later stages of optimization. Also note that FastDOG's iteration complexity is linear in the size of the BDD-problem decomposition, hence one iteration is comparatively cheap. Moreover, it is easily parallelizable, explaining FastDOG's and our speedup as compared to Gurobi.
>
> ### 9. I assume that in line 123 should be replaced with according to the line 10 in Alg. 1.
> In FastDOG, the value of $\omega$ was actually equal for all dual variables and so the authors treated it as a scalar. We will resolve this ambiguity in the final manuscript.

---

> > ### Comment · Reviewer_jrD7 · 2022-08-09
> > **Response**
> >
> > Thanks for answering questions! Now, the proposed approach becomes technically more clear and the presentation is more consistent. The proper context and specialisation of the proposed approach to solver construction is important and I like the presented results during discussion with reviewer ehgi. Please, add the respective comments in the manuscript about the expected workflow and results from running your solver compare with some specialised or commercial competitors.

---

> > > ### Author Response · Authors · 2022-08-09
> > > **Additions in the final manuscript**
> > >
> > > We thank the reviewer for detailed feedback. We will add more context and discussion about specialized solvers in the final version.
> > >
> > > _Regards,_
> > >
> > > _Paper 3276 authors_

---

> ### Author Response · Authors · 2022-07-29
> **Response to weakness pointed out by reviewer jrD7**
>
> Thanks for detailed reading and review. We breakdown our reply in two parts due to lack of space. A minor clarification: we do not have a separate model for predicting the non-parametric update. Instead a GNN predicts all three vectors $\alpha$, $\omega$ and $\theta$ simultaneously as in Line 4 of Alg. 3.
>
> ### Weakness 1: Hand-crafted GNN features
> The current state of Alg. 1 is completely determined by the current set of dual variables $\lambda$ which can easily be encoded as edge features of the GNN. We additionally encode some hand-crafted features like subgradients or min-marginals allowing the GNN to construct well-performing algorithms like subgradient ascent or min-marginal averaging. Therefore they were a natural choice for input into our GNN. The same holds true for keeping a history of intermediate states for DOGE-M. Similar history information is used in accelerated gradient methods [NE83] and heavy ball methods [PO64] to accelerate first order methods. Also other features were quite natural, including the ILP description, which was used in [35]. In general we expect our solver to become better once other informative features are provided, for which the optimization literature might be a good source.
>
> [PO64] - "Some methods of speeding up the convergence of iteration methods", B. Polyak, USSR Computational Mathematics and Mathematical Physics, 1964.
>
> [NE83] - "A method of solving a convex programming problem with convergence rate $O(1/k^2)$”, Y. Nestrov, Soviet Mathematics Doklady, 1983
>
> ### Weakness 2: Faster than Gurobi, insights
> Actually our GNN is pretty lightweight, containing less than $10k$ parameters. The GNN predicts parameters for the solver once only after several iterations of the solver (see Table 1, column $T$, subcolumn $test$) thereby reducing the computational burden. Second, backpropagation and training is not done during test time, making our approach fast at inference time. Third, our approach can utilize massive parallelism offered by GPUs, in contrast to Gurobi and other solvers which can only run on CPU.

---

> > ### Comment · Reviewer_jrD7 · 2022-08-09
> > **Response**
> >
> > Thanks for the comments about the aforementioned weaknesses!
> > The manuscript can be improved by adding more discussion related to this comments. In particular, motivation for the used features for GNN can be briefly added and distribution of runtime over the GNN operations, optimizer steps makes clear the overhead from additional operations and gain in reducing optimizer-only runtime.
> > In addition, the usage of history to accelerate methods is very fruitful idea but it is unclear how the GNN transforms the history and why such transformation speeds up the convergence. So, the idea is clear, but it requires additional investigation.

---

> > > ### Author Response · Authors · 2022-08-09
> > > **Changes for final version**
> > >
> > > We thank the reviewer for suggesting these changes, we will add all of the above points in the final version. We agree that currently it is not clear what part of history the LSTM maintains to speed-up convergence. This can be a good direction for future investigation.
> > >
> > > _Regards,_
> > >
> > > _Paper 3276 authors_

---

### Official Review · Reviewer_TgLG · 2022-07-11

**Rating:** 4
**Confidence:** 2
**Soundness:** 2 fair
**Presentation:** 2 fair
**Contribution:** 2 fair

**Summary:**

The authors present DOGE, a learning-based method to solve relaxations to integer programs that exploits GPU parallelism.
Specifically, DOGE builds on a method based on Lagrangian decomposition, FastDOG, and proposes to improve its update step either by optimizing over its parameters, or by learning the update direction via a GNN.
DOGE is shown to improve on FastDOG on 4 benchmarks.

**Questions:**

- Could the authors include the omitted negative results mentioned in lines 278-285?

**Limitations:**

The authors adequately and honestly address the limitations of their work.

**Strengths And Weaknesses:**

The paper is well written and easy to follow. While the results appear to be convincing on the selected benchmarks, DOGE is an incremental improvement on FastDOG, which introduced the GPU-friendly dual formulation. I believe the originality of DOGE to be limited as it only proposes to learn the update steps of FastDOG, while keeping the same formulation. Given the extent of the improvement over FastDOG over the selected benchmarks, and lack thereof in the experiments mentioned in lines 278-285, I am not sure that the cost associated to training pays off in this context.

---

> ### Author Response · Authors · 2022-07-29
> **Response to reviewer TgLG**
>
> ### Incremental improvement:
> We think that our work is indeed novel in the following key aspects:
> - We show how to backpropagate through FastDOG (Alg. 2) and with additional derivations in the supplementary material. This allows us to obtain the first fully differentiable solver for convex relaxations of combinatorial problems and demonstrate that machine learning can have an impact in this area of optimization solver development.
> - Our approach is unsupervised and self-sufficient, in contrast to methods that rely on ground truth supervision [35], imitation learning [L2C] or reinforcement learning approaches [43, 55] that do not need supervision but heavily rely on classical algorithms for core parts of their computation.
> - The improvement of DOGE over FastDOG is substantial, since typically closing the dual gap, which DOGE largely does, is substantially harder than reaching an acceptable, but suboptimal dual solution, which FastDOG delivers. In optimization, solving the last few percent of a problem typically takes the majority of the time (for example see Figure 2b, 2c in [43]).
>
> [L2C] - Paulus, Max B., et al. "Learning to Cut by Looking Ahead: Cutting Plane Selection via Imitation Learning." ICML 2022.
>
> ### Omitted results
> We will provide these results in a final version of the paper. We would like to note that our training does not lead to a solver that is worse than FastDOG, though.

---

> > ### Comment · Reviewer_TgLG · 2022-08-08
> > **Response**
> >
> > I sincerely thank the authors for their response.
> > While I appreciate the empirical effectiveness of DOGE over FastDOG, I still stand by my concerns on the improvement being incremental, especially given the cost of training (see discussion with reviewer ehgi).

---

> > > ### Author Response · Authors · 2022-08-08
> > > **Incremental improvement over FastDOG**
> > >
> > > Thanks for the response. Even if our method is trivial and incremental (which we believe is not) we would kindly refer the reviewer to inspect our results in Table 3. We always get better objective values than FastDOG with **several times better** value of relative dual gap integral $g_I$ [7]. Specifically:
> > >
> > > _Cell tracking:_ Improvement by $3 \times$
> > >
> > > _Graph matching:_ Improvement by $45 \times$
> > >
> > > _Independent set:_ Improvement by $70 \times$
> > >
> > > _QAPLib:_ Improvement by $26 \times$
> > >
> > > Note that FastDOG will not be able to achieve better objectives values than ours **even if it is given unlimited time** because it can get stuck in fixed points.

---

### Official Review · Reviewer_ehgi · 2022-07-24

**Rating:** 3
**Confidence:** 3
**Soundness:** 2 fair
**Presentation:** 2 fair
**Contribution:** 2 fair

**Summary:**

Authors propose an extension of the work A.Abbas and P.Swoboda FasdDOG, 2022. New algorithm includes graph network, which allows to learn parameters for non-parametric update and parametric step of FastDOG, 2022.

**Questions:**

1. How the neural network is trained? During the optimization process, or separately?
2. For me it is unclear, why in Table 1 training time is in hours and significantly large compared to Figure 2.
3. If training and performing optimization are separate processes, training time should be included in Figure 2.
4. For table 2 it is also interesting to see the comparison in the number of iterations to understand the complexity of one iteration
5. Line 246. As for me, it is unfair to fix parameters in Alg. 1 for the CT dataset. It creates a fake impression of the proposed approach.

**Limitations:**

-

**Strengths And Weaknesses:**

Strengths:
1. Modification of FastDOG algorithm, which allows for:
    - graph NN for learning parameters
    - GPU parallelization
    - non-parametric step

Weaknesses:
1. The resulting algorithm includes a non-parametric step, but there are no theoretical guarantees for the non-parametric step.
2. Experiments part seems to be a bit unclear for me. Questions are listed in the corresponding section.
3. Experiments show that on 3 out of 4 datasets Gurobi shows better results and on two of them Gurobi does it in approximately the same time.
4. The code is not attached.

---

> ### Author Response · Authors · 2022-07-29
> **Response to reviewer ehgi**
>
> ### Strengths:
> In addition to the strengths pointed out by the reviewer we would like to add the following:
>
> 1. Our test instances are significantly larger than training ones (ref. Table 1). For example we test on $20\times$ larger instances in QAPLib dataset. This is already a difficult task as also stated in Sec. 6.3 of the survey [6].
> 2. We do not need ground truth solutions for training due to our unsupervised loss.
> 3. We show how to backprop through FastDOG for end-to-end training.
> 4. Please also see our response to reviewer TgLG.
>
> ### Weakness 1: Theoretical guarantees:
> While it is true that the non-parametric update step does not provide convergence guarantees, we provide the GNN subgradients and min-marginals as features, from which it is easily possible to construct a converging algorithm. Also, many state of the art optimization algorithms trade theoretical guarantees for empirical convergence, e.g. heuristic step sizes in subgradient schemes [SG] or aggressive combinatorial moves in max-flow [BK].
>
> [SG] - Bazaraa, Mokhtar and Sherali, Hanif: "On the choice of step size in subgradient optimization" in European Journal of Operational Research 1981
>
> [BK] - Boykov, Yuri and Kolmogorov, Vladimir: "An Experimental Comparison of Min-Cut/Max-Flow Algorithms for Energy Minimization in Vision" in TPAMI 2004
>
> ### Weakness 2: Questions regarding experiments:
> 1. **How the neural network is trained? During the optimization process, or separately?** The neural network is trained with the optimization problem embedded inside it. For updating neural network weights we backprop through this optimization problem by Alg. 2. For testing, the neural network weights are frozen so backprop is not required at test time.
> 2. **For me it is unclear, why in Table 1 training time is in hours and significantly large compared to Figure 2.** In Table 1 we report our training setting. Figure 2 contains our main results on test instances computed using an already trained GNN.
> 3. **If training and performing optimization are separate processes, training time should be included in Figure 2.**
> We follow the conventional way to report results of machine learning methods for combinatorial optimization and measure test time performance, while training time etc. is regarded as a secondary measure. See for example the recent competition of [ML4CO] and the well-cited paper in the area [35]. The reason is that training is done only once and then the model can be used subsequently without incurring the training time overhead. Note also that our training scheme is already much faster than other approaches using machine learning for combinatorial optimization. For example, the work for learning to solve travelling salesman problems [54] requires 4 days for training and more for ground truth generation.
> 4. **Complexity of one iteration in Table 2**
> For the ablation study in Table 2 per iteration runtime of all learned approaches is almost similar ($762ms$). The reason is that the GNN is invoked only after several iterations (see column $T$, $test$ in Table 1) of the solver and so using a GNN or not (w/o GNN) does not change runtime by much. Also the non-parametric update is performed only once whenever the GNN is invoked and thus does not impact total runtime much. The runtime of one iteration of DOGE is empirically $1.2x$ the cost of one FastDOG iteration (due to variable $\alpha$ and $\omega$ parameters). We provide the number of iterations for each test instances in the appendix. Note that FastDOG's iteration complexity (and ours) is linear in the size of the BDD problem decomposition.
> 5. **Fix parameters in Alg. 1 for CT dataset**
> For the CT dataset we only train on two training samples which might be too few for generalization if we predict all parameters ($\alpha$, $\omega$, $\theta$). Note that we do still predict the parameters $\theta$ for the non-parametric update (eq. 2) which allows us to outperform FastDOG by a large margin.
>
> [ML4CO] - https://www.ecole.ai/2021/ml4co-competition/
>
> ### Weakness 3: Comparison to Gurobi
> While Gurobi reaches slightly better final objectives on most datasets, actually we are very close to optimum already and the differences are on the order of $10^{-3}$ w.r.t. the relative dual gap (ref. Figure 2). On the other hand our any-time performance is orders of magnitude better, which means that when the algorithm is stopped early our results would be significantly better. Also the time as well as objective axes are logarithmic, hence the time differences between Gurobi and our approach are actually significant as well. In addition, beating Gurobi, which is a closed source commercial solver developed by a large team of dedicated researchers is typically not achieved by academic papers. Lastly we quote from a recent related survey paper [10] (pg14) "Optimality is desired, but quickly finding a good solution is the priority".
>
> ### Weakness 4:  Code is not attached
> Please see the overall comment.

---

> > ### Comment · Reviewer_ehgi · 2022-08-04
> > **Response**
> >
> > Dear authors, thank you for the response!
> >
> > Strengths. Well, isn't everything you mentioned a part of "graph NN for learning parameters"?
> >
> > 1-3. Can the neural network be pretrained once for different optimization problems? Can a neural network pretrained on one dataset show some improvements on another? It seems that to solve the new optimization problem neural network has to be trained again, which takes from 4 to 48 hours.
> >
> > As far as I understand, by training neural network separately you give the optimizer some "prior" knowledge about the dataset and optimization problem. So there is no surprise that the proposed approach improves results.
> >
> > I still don't think that it is fair to separate training from the optimization process without showing that only once trained NN can behave well on different optimization problems.

---

> > > ### Author Response · Authors · 2022-08-05
> > > **About generalization and accounting training time**
> > >
> > >
> > > ### 1. To solve the new optimization problem neural network has to be trained again
> > >
> > > a. It is still important to have dataset-specific solvers as they can exploit the problem structure. For practitioners it is common to solve problems of same type over and over again. For example take example of a microbiologist who would like to track cells in a video whenever new data is acquired. To speed-up optimization, the default course of action is to first develop application specific solvers taking considerable time and human effort. On the other hand **training our solver is still less time consuming than developing an application specific solver**. Note that FastDOG already was already competitive with dataset specific solvers especially in early stages of optimization but got stuck in suboptimal fixed points. Our method alleviates this issue while also achieving faster convergence.
> > >
> > > b. Nonetheless, the problem of generality in machine learning is ubiquitous and it would be good to have application agnostic trained solver. However, **our contribution is not in learning a 'general' AI agent nor in zero-shot learning**. We use standard GNN with only one message passing layer and standard gradient based NN optimizer. In the domain of learning to optimize we would also point out to Sec. G of ICML'22 spotlight paper [L2C] where the authors also comment on lack of generalization of their method.
> > >
> > > [L2C] - Learning to Cut by Looking Ahead: Cutting Plane Selection via Imitation Learning. Paulus et. al. ICML 2022.
> > >
> > > ### 2. You give the optimizer some "prior" knowledge ... No surprise that the proposed approach improves results
> > >
> > > Even if we agree with this statement it does not imply that:
> > >
> > > a. Our work will not benefit the research community and practitioners.
> > >
> > > b. The work is not novel.
> > >
> > > ### 3. I still don't think that it is fair to separate training from the optimization process
> > > For the sake of discussion even if we were to perform this comparison our method might still have some advantages:
> > >
> > > **a. 'Fair' comparison with Gurobi:**
> > > Our method is competitive with Gurobi even if we take this type of fairness into consideration for large datasets. For example for our largest dataset QAPLib which takes most amount of training time (2 days). Evaluating our solver on all test instances takes around 8 hours. Thus our solver spends 2.3 days in total (training + inference) on QAPLib dataset. We ran the commercial solver Gurobi with a time limit of 1 hour for each test instance and it terminated in total of 31.8hours (1.3 days) with **much worse lower bounds**. Specifically as in Table 3, average objective value for Gurobi was $0.9 \times 10^6$ whereas for our method $14.5 \times 10^6$. We can add one more experiment in the final version where we give same amount of time budget to Gurobi as in our training + testing.
> > >
> > > Lastly, it might be possible to speed-up our training process further by solving all ILPs in a training batch in parallel and caching strategies. We currently solve each ILP sequentially, parallelizing this requires considerable engineering effort which is beyond the scope of this work.
> > >
> > > **b. 'Fair' comparison with FastDOG:**
> > > Since FastDOG can get stuck in sub-optimal fixed points far away from the optimum it will still not be able to achieve the dual objectives even if it is given unlimited amount of time.
> > >
> > > Regards,
> > >
> > > Paper 3276 authors

---

> > > > ### Comment · Reviewer_ehgi · 2022-08-09
> > > > **.**
> > > >
> > > > Dear authors, thank you for the response!
> > > >
> > > > **1**
> > > >
> > > > "Training our solver is still less time consuming than developing an application specific solver"
> > > > According to experiments, we can use Gurobi on most datasets and it will outperform DOGE in a comparable time.
> > > >
> > > > "On the other hand training our solver is still less time consuming than developing an application specific solver. Note that FastDOG already was already competitive with dataset specific solvers especially in early stages of optimization but got stuck in suboptimal fixed points."
> > > > I don't see experiments comparing DOGE with application-specific solvers. According to experiments from the FastDOG paper can't you also say that Gurobi is competitive with application-specific solvers?
> > > >
> > > > **3**
> > > > " We can add one more experiment in the final version where we give same amount of time budget to Gurobi as in our training + testing."
> > > > Thank you, I'd like to see a such experiment with all considered algorithms.
> > > >
> > > >
> > > > **To sum up**
> > > > After discussion with the authors, I still have the same concerns:
> > > > - Train-test separation
> > > >    I am still not convinced that training network and optimizing are separate processes in optimization. Even the authors provided an example of the repetitive task it is only a small class of problems. Moreover, DOGE outperforms commercial solver Gurobi only on one dataset in terms of accuracy.
> > > > - Practicality
> > > >   In my opinion due to the large training time (4 to 48 hours) DOGE can not become a commercial optimizer. On Cell Tracking, Graph Matching, Independent Set Gurobi achieves better accuracy in less time.
> > > > - Theoretical Contribution
> > > >  One theoretical results claim that you can backpropagate, the other is taken from FastDOG. I agree with the authors that " ...many state of the art optimization algorithms trade theoretical guarantees for empirical convergence". So theoretical results can't be seen as strengths nor as weaknesses.
> > > >
> > > >
> > > > So, in the end, we have a dataset-specific solver that shows some improvements only on one dataset.
> > > > - The main idea is to use NN to learn hyperparameters, which is very interesting. However, Gurobi outperforms it in terms of accuracy on 3 datasets out of 4.
> > > > - Improvements in the speed of convergence are because the neural network gives a priori knowledge about the dataset. If we do not separate training and optimization, DOGE is much more time-consuming on 3 datasets out of 4.
> > > > - Practicality is doubtful due to the large training time, and generalization ability is unclear. Solver is dataset-specific.
> > > > - For CT dataset parameters are fixed. How they are tuned? Fixing hyperparameters in the approach where they should be learned creates a negative impression.
> > > > - On the QAPLib dataset DOGE outperforms other approaches. Authors claim that given the same amount of time DOGE outperforms Gurobi.
> > > > - DOGE outperforms FastDOG in terms of accuracy. However, DOGE needs more time due to the training process.
> > > >
> > > > Maybe the authors need to refocus the paper on complex datasets like QAPLib (maybe with repetitive problems), where the superiority of DOGE will be clear.

---

> > > > > ### Author Response · Authors · 2022-08-09
> > > > > **Training time, practicality, Repetitive problems**
> > > > >
> > > > > We thank the reviewer for their response.
> > > > >
> > > > > ## Final accuracy comparison with Gurobi:
> > > > > Final accuracy is only one metric but other metrics are also useful and practical. Concretely, even though given unlimited time Gurobi outperforms DOGE, reasonably close values to the optimum are also considered good for practical purpose. This is because the inputs to the ILP are also not known with complete certainty and thus the practitioners require good enough solutions but in less time. For example quoting the very recent graph matching [benchmark](https://vislearn.github.io/gmbench/results/) _"we consider solutions within a 0.1% range to known optimum as optimal"_. With respect to this metric and the metric of relative dual gap integral [7] we outperform Gurobi even though it is a commercial solver with much resources at disposal.
> > > > >
> > > > > ## Comparison with specialized solvers:
> > > > > 1. Note that specialized solvers (and FastDOG) mostly struggle with solving the LP relaxation and thus their lower bounds are worse. On the primal task these solvers are actually better than Gurobi if one needs to compute good (but not necessarily optimal) solutions in a limited amount of time.
> > > > >
> > > > > 2. We report comparison of DOGE with best known specialized solvers as follows:
> > > > >
> > > > > ### _Cell tracking_:
> > > > >
> > > > > | Solver type                   | $E (\times 10^8)$ | Runtime(seconds) |
> > > > > |---|---|---|
> > > > > | Specialized solver [24] |  -3.866 | 1673 |
> > > > > |  DOGE                           | **-3.854** |  1380   |
> > > > > |  DOGE-M                      |   **-3.854** | **730**      |
> > > > > Thus the specialized solver [24] does not get a lower bound $E$ close to ours and seems to be stuck (by inspecting the solver log).
> > > > >
> > > > > ### _Graph matching_:
> > > > > For graph matching dataset we obtain results of specialized solvers from very recent work [GMB] which is to be presented at ECCV'22. We find **the best specialized solver for each instance** and report the results below:
> > > > >
> > > > > | Solver type                   | $E$ |
> > > > > |---|---|
> > > > > | Best spec. solver after 10 seconds  | -48442 |
> > > > > | Best spec. solver after 100 seconds  | -48441 |
> > > > > | Best spec. solver after 300 seconds  | -48441 |
> > > > > |  DOGE after 17 seconds                                      |   **-48439** |
> > > > > |  DOGE-M after 21 seconds                           | **-48436** |
> > > > >
> > > > > Thus we actually achieve better lower bounds than the best specialized solver and still with less runtime.
> > > > >
> > > > > ## Practicality is doubtful:
> > > > > As shown above for both cell tracking and graph matching problems there is much interest in solving these problems (and particularly the LP relaxations) faster than Gurobi [24, GMB]. This has motivated the need for specialized solver development separately for each dataset. We forego this need for spending large amount of human effort and develop a general purpose solver which can trained for each problem type in much lesser amount of time than developing specialized solver.
> > > > >
> > > > > [GMB] - A Comparative Study of Graph Matching Algorithms in Computer Vision. Haller et. al, ECCV 2022.
> > > > >
> > > > > ## Refocus on complex datasets with repetitive problems:
> > > > > 1. To compete w.r.t this metric one can easily inflate the testing set by acquiring more data.
> > > > > 2. This is the reason why we separate training time and testing time. This already demonstrates to possible users having repetitive problems that once a model is trained (paying a one time cost) it allows fast inference.
> > > > >
> > > > > ## CT dataset:
> > > > > Given an instance with $n$ many dual variables the GNN stills predicts parameters for non-parameteric update $\theta \in R^n$. Rest of the two parameters $\alpha$ and $\omega$ are fixed to the values used in FastDOG. Note that for CT dataset the parameter space is still large because $n = 28 \times 10^6$ (ref. Table 1) which is even larger than the size of our GNN's trainable parameters ($10 ^ 4$).
> > > > >
> > > > > ## DOGE can not become a commercial optimizer:
> > > > > We cannot and do not strive for this, as we do not have access to amount of resources (personnel, funding, hardware). Our solver is a limited academic effort. Still we show that there are cases where our solver can be beneficial as compared to the commercial and specialized solvers.
> > > > >
> > > > > _Regards,_
> > > > >
> > > > > _Paper 3276 authors_

---

### Author Response · Authors · 2022-07-29
**Code availability**

We provide anonymous link of our implementation here: https://anonymous.4open.science/r/DOGE-Train-C089/. Efficient CUDA Implementation of Alg. 1 and its backpropagation (Alg. 2) can be found at [bdd_cuda_learned_mma.cu](https://anonymous.4open.science/r/DOGE-Train-C089/external/BDD/BDD-main/src/bdd_cuda_learned_mma.cu), GNN model is defined at [model.py](https://anonymous.4open.science/r/DOGE-Train-C089/model/model.py). We will also provide publicly all code and pretrained models upon acceptance of the paper.

---

### Meta-Review · Area_Chair_TUJH · 2022-08-27

**Recommendation:** Reject
**Confidence:** Less certain

**Metareview:**

The presented paper introduces DOGE-Train method that targets discrete optimization problems. It allows finding solutions for discrete problems utilizing GPUs. This is achieved by pre-training on smaller size instances and then hoping it would also generalize for larger instances that are coming from a similar family of problems. Overall, the idea is related to FastDOG but has some improvements.

Despite showing promising results, there are still a few concerns raised by reviewer "ehgi". Also, I am not sure if the NeurIPS would be the best fit for this paper.



**Award:**

No

---

### Decision · Program_Chairs · 2022-09-14

Reject